# Association of Objects May Engender Stereotypes: Mitigating Association-Engendered Stereotypes in Text-to-Image Generation

**Junlei Zhou[1], Jiashi Gao[1], Xiangyu Zhao[2], Xin Yao[3], Xuetao Wei[1]***
[1]Southern University of Science and Technology, [2]City University of Hong Kong
[3] Lingnan University
`{zhoujl2023,12131101}@mail.sustech.edu.cn, xy.zhao@cityu.edu.hk`
`xinyao@ln.edu.hk, weixt@sustech.edu.cn`

## Abstract

Text-to-Image (T2I) has witnessed significant advancements, demonstrating superior performance for various generative tasks. However, the presence of stereotypes in T2I introduces harmful biases that require urgent attention as the T2I technology becomes more prominent. Previous work for stereotype mitigation mainly concentrated on mitigating stereotypes engendered with individual objects within images, which failed to address stereotypes engendered by the association of multiple objects, referred to as *Association-Engendered Stereotypes*. For example, mentioning "black people" and "houses" separately in prompts may not exhibit stereotypes. Nevertheless, when these two objects are associated in prompts, the association of "black people" with "poorer houses" becomes more pronounced. To tackle this issue, we propose a novel framework, `MAS`, to Mitigate Association-engendered Stereotypes. This framework models the stereotype problem as a probability distribution alignment problem, aiming to align the stereotype probability distribution of the generated image with the stereotype-free distribution. The MAS framework primarily consists of the *Prompt-Image-Stereotype CLIP* (*PIS CLIP*) and *Sensitive Transformer*. The *PIS CLIP* learns the association between prompts, images, and stereotypes, which can establish the mapping of prompts to stereotypes. The *Sensitive Transformer* produces the sensitive constraints, which guide the stereotyped image distribution to align with the stereotype-free probability distribution. Moreover, recognizing that existing metrics are insufficient for accurately evaluating association-engendered stereotypes, we propose a novel metric, *Stereotype-Distribution-Total-Variation* (*SDTV*), to evaluate stereotypes in T2I. Comprehensive experiments demonstrate that our framework effectively mitigates association-engendered stereotypes.

## 1   Introduction

Text-to-Image (T2I) (Nichol et al. [2021], Rombach et al. [2022], Ramesh et al. [2022], Saharia et al. [2022]), based on the diffusion model, has achieved significant breakthroughs in image generation, showing potential for various downstream tasks such as image creation, editing (Brooks et al. [2023]), etc. In addition, the scale of T2I applications has expanded impressively. For example, the open-source stable diffusion project (Rombach et al. [2022]) has gained the favor of more than 10 million users. As of August 2023, the number of images created through T2I has surpassed 15 billion[†].

---

*Corresponding author.
[†]https://journal.everypixel.com/ai-image-statistics

38th Conference on Neural Information Processing Systems (NeurIPS 2024).

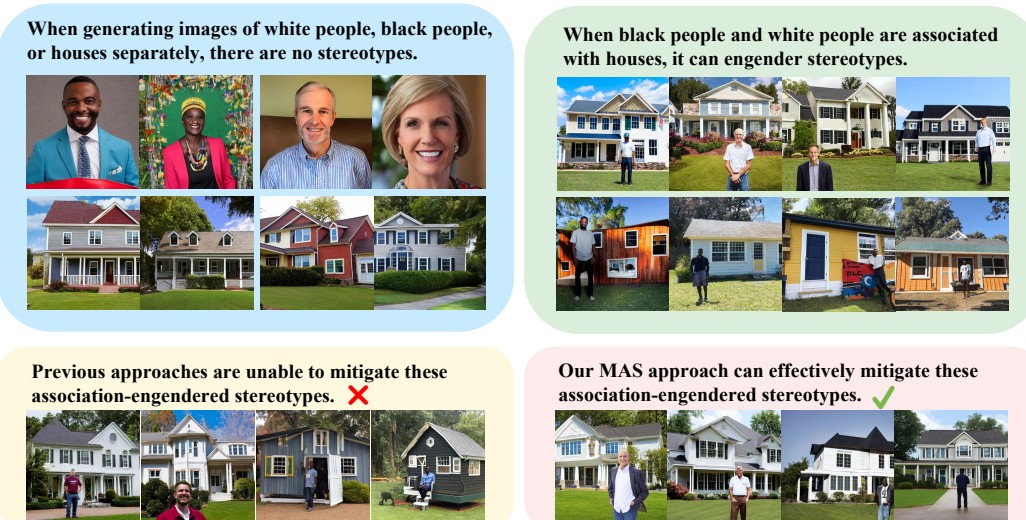

Figure 1: Original T2I models can generate stereotypes when given prompts involving multiple objects; when the prompts are "*a photo of black/white people*" or "*a photo of a house*", they do not engender stereotypes. However, if the prompt is "*a photo of black/white people and a house*", the model may engender stereotypes that the *white people's* house is better than the *black people's* house. It is noted that previous mitigation approaches have yet to mitigate these association-engendered stereotypes. Our approach, MAS, demonstrates the capability to effectively mitigate these stereotypes.

As images generated by diffusion models become more prevalent, mitigating the stereotypes they engender is increasingly critical (Seshadri et al. [2023], Luccioni et al. [2024]).

The stereotype in social psychology is a mental construct of a particular type of person, object, or thing, often based on observing individuals who fit that type. Previous work explored sensitive attributes (such as gender, race, region, religion, and age) and corresponding stereotypes present in T2I diffusion models (Wang et al. [2023a], Esposito et al. [2023], Bianchi et al. [2023]). For instance, when the prompt is "*a photo of a politician*", the object (politician) in the image generated by the T2I diffusion model always appears *male* in the sensitive attribute of gender. Previous works (Shen et al. [2024], Kim et al. [2023], Esposito et al. [2023]) on stereotype mitigation have been limited to a single object, referred to as *Non-Association-Engendered stereotypes* (e.g., only mitigating the stereotype problem in the occupation or gender dimension), and cannot effectively address stereotypes involving the association of multiple objects, referred to as *Association-Engendered Stereotypes*. For example, as shown in Figure 1, when the prompt is "*a photo of a white people and a house*" versus "*a photo of a black people and a house*", T2I diffusion models often generate images where the houses associated with white people appear better than those associated with black people, reflecting an association-engendered stereotype. Therefore, mitigating such association-engendered stereotypes is highly desired for T2I models.

In this paper, our contributions are as follows: ❶ To the best of our knowledge, we take the first step towards addressing association-engendered stereotypes in T2I models. We define the stereotype problem in the T2I diffusion model as a probability distribution alignment problem, aiming to align the stereotype probability distribution of the generated image with the stereotype-free distribution. ❷ We present a novel framework MAS to mitigate the association-engendered stereotypes, which primarily consists of the *Prompt-Image-Stereotype CLIP* (*PIS CLIP*) and *Sensitive Transformer*. The *PIS CLIP* learns the association between prompts, images, and stereotypes, which can construct the mapping of prompts to stereotypes. The *Sensitive Transformer* produces the sensitive constraints, which guide the stereotyped image distribution to align with the stereotype-free probability distribution. ❸ Given the insufficiency of existing metrics for accurately evaluating association-engendered stereotypes, we propose a novel metric for evaluating stereotypes in T2I: *Stereotype-Distribution-Total-Variation* (*SDTV*). ❹ We conduct a comparative evaluation of our stereotype mitigation approach on five popular T2I diffusion models and six advanced stereotype mitigation approaches in T2I. Extensive experiments demonstrate the superiority of our approach in effectively mitigating both non-association-engendered and association-engendered stereotypes.

Table 1: Stereotype categories and prompt word templates

| Abbreviation | Stereotype Categories | Prompt |
|---|---|---|
| S-O & S-SA | Single Object with a Single Sensitive Attribute | A photo of [OBJECT]. |
| S-O & M-SA | Single Object with Multiple Sensitive Attributes | A photo of [SA] [OBJECT]. |
| M-O & S-SA | Multiple Objects with a Single Sensitive Attribute | A photo of [OBJECT 1] and [OBJECT 2]. |
| M-O & M-SA | Multiple Objects with Multiple Sensitive Attributes | A photo of [OBJECT 1] and [OBJECT 2]. |

## 2 Related work

**Stereotypes in T2I models.** Although the current image quality and capabilities generated by T2I are gradually approaching maturity, it is worth noting that various biases and stereotypes may still arise from neutral prompt words (Zhou et al. [2024]). Fraser et al. [2023] and Wan and Chang [2024] found that text-generated image models often engender stereotypes of gender, race, and demographics in sociology. These biases can manifest in multiple aspects, such as occupations (Bianchi et al. [2023], Cho et al. [2023], Luccioni et al. [2023]), objects (Mannering [2023]) and adjectives (Luccioni et al. [2023], Naik and Nushi [2023]). Additionally, T2I diffusion models may exhibit gender bias when depicting interactions between two or more people. Based on the above findings, previous researchers proposed stereotype detection frameworks such as Language Agent (Wang et al. [2023a]) and T2IAT detection (Wang et al. [2023b]). These research results provide practical tools for identifying stereotypes and lay a solid foundation for further mitigating stereotypes.

**Stereotypes mitigation in T2I models.** Existing approaches to mitigate stereotypes in T2I diffusion models have effectively mitigated some of the stereotypes but still have some limitations. Friedrich et al. [2023] randomly introduced additional text clues by identifying known occupations in the prompts to ensure a fair distribution of the generated images. However, this approach may ineffectively address biases related to occupations that are not predefined. Kim et al. [2023] developed a de-stereotyping loss function and adjusted specific parameters of the soft prompts to balance the sensitive attributes of the generated images but only achieved de-stereotyping in the T2I model containing occupational prompts. Chuang et al. [2023] proposed a technique that involves projecting biased directions in text embeddings and debiasing by utilizing text data with a calibrated projection matrix. Fine-tuning the diffusion model by modifying the capabilities of the pre-trained model has also proven to be an effective strategy. Inspired by Gal et al. [2023], Zhang et al. [2023], Brooks et al. [2023], Dai et al. [2023], Shen et al. [2024] fine-tuned the sampling process of the diffusion model and used an optimized loss function to align the T2I model with fairness principles directly. Moreover, He et al. [2024] utilized an iterative distribution alignment approach to align biased distributions with a unified distribution. Schramowski et al. [2023] built a Safe Latent Diffusion (SLD) capable of suppressing sensitive image portions during diffusion. Previous researchers adjusted the original T2I model and had to consider the image quality and mitigate stereotypes in the original T2I model. They mainly focused on mitigating non-association-engendered stereotypes in the T2I model (such as those engendered by occupation/gender), so it is unable to mitigate the stereotypes generated when multiple objects are associated effectively.

## 3 Our framework: mitigate association-engendered stereotypes (`MAS`)

In this section, we construct a mathematical model of stereotypes in T2I. Based on this model, we propose a novel metric, *Stereotype Distribution Total Variation* (*SDTV*), for evaluating stereotypes. Then, we model the stereotype mitigation problem as a probability distribution alignment problem and build a new stereotype mitigation framework, Mitigate Association-engendered Stereotypes (`MAS`). Finally, we train *PIS CLIP* and *Sensitive Transformer* models to construct sensitive constraints, which guide the stereotype probability distribution to align with the stereotype-free probability distribution.

### 3.1 Modeling stereotypes in T2I

Inspired by Shen et al. [2024], we denote the object of the image as $x$ and the object's sensitive attributes as $s$ (such as *gender, race, and adjective*). We introduce the probability distribution function (PDF) $P(s = v(s)|x)$ to quantify the likelihood of the image's object exhibiting specific sensitive attributes, where $v(s)$ represents the value of the sensitive attribute. For instance, when

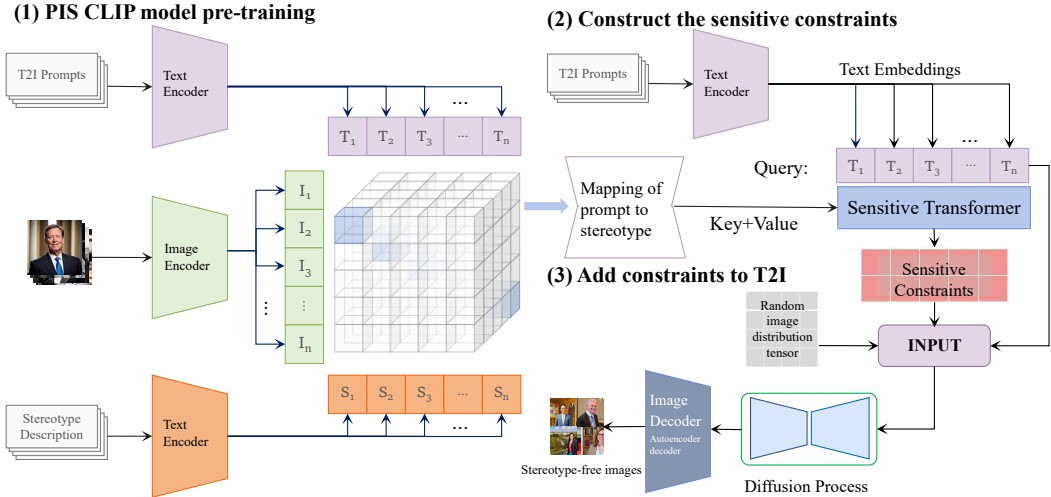

**(1) PIS CLIP model pre-training**

**(2) Construct the sensitive constraints**

**(3) Add constraints to T2I**

Figure 2: Overview of our proposed MAS. (1) In the PIS model pre-training stage, we adjust the traditional CLIP structure from the original "text-image" two-dimensional mapping to a "text-image-text" three-dimensional mapping, thereby obtaining a mapping of prompts to stereotypes. (2) In constructing sensitive constraints, we propose a *Sensitive Transformer* based on the transformer structure to construct sensitive constraints for each prompt. (3) In the stage of adding sensitive constraints to the T2I, we embed the sensitive constraints into the T2I diffusion model to guide the generation of stereotype-free images.

the generated image's object $x$ is *person* and $s$ is *gender*, $v(s)$ could be *male* or *female*. The PDF represents the probability of a person appearing as male or female in the sensitive attribute of gender. Furthermore, to distinguish the difference between association-engendered stereotypes and stereotypes explored by previous work (Kim et al. [2023], He et al. [2024], Shen et al. [2024]), we categorize stereotypes into four distinct categories (Table 1) based on the association between objects and sensitive attributes: ❶ a single object with a single sensitive attribute (S-O & S-SA, non-association-engendered stereotypes), ❷ a single object with multiple sensitive attributes (S-O & M-SA, non-association-engendered stereotypes), ❸ multiple objects with a single sensitive attribute (M-O & S-SA, association-engendered stereotypes), ❹ multiple objects with multiple sensitive attributes (M-O & M-SA, association-engendered stereotypes). Here, we utilize a joint PDF to define categories with multiple objects with multiple sensitive attributes.

$$f(x_1, x_2, \dots) = P\big(s_{x_1} = v(s_{x_1}), s_{x_2} = v(s_{x_2}), \dots | x_1, x_2, \dots\big), \tag{1}$$

where $x_1$ and $x_2$ represent the two objects in the image. For instance, in Figure 1, $x_1$ corresponds to a *people*, and $x_2$ corresponds to a *house*. The variables $s_{x_1}$ and $s_{x_2}$ denote the *race* and *house's description*. See Appendix A for more PDF definitions of other stereotype categories.

## 3.2 A new metric to evaluate stereotypes: *SDTV*

Stereotypes engender when a sensitive attribute in an image consistently presents one or a few states in most instances (Naik and Nushi [2023]). Therefore, Saravanan et al. [2023] and Wang et al. [2023a] have utilized various metrics to evaluate the stereotype from the perspective of the overall proportion of different subgroups. However, when evaluating association-engendered stereotypes, the proportion of subgroups in the population cannot accurately evaluate association-engendered stereotypes. To address this shortcoming, we propose a new stereotype evaluation metric using total variation distance based on the probability distribution model of stereotypes in T2I.

Consider a T2I model, denoted as $\mathcal{G}$, which generates an image, and $x$ is an object of the image. We define a set $S_x$ denotes that object $x$ contains multiple sensitive attributes such that $S_x = \{$gneder, race, region, $\dots\}$. We assign the sensitive attributes as $v(s_x)^1, v(s_x)^2, \dots, v(s_x)^{\mathcal{Y}}$ for each sensitive attribute $s_x$, $\forall s_x \in S_x$ (e.g. $v(s_x) = $ *female*, $v(s_x) = $ *black*, $\dots$). For an object (i.e., $x = $"*doctor*" or $x = $"*nurse*"), the stereotype-free T2I model should generate images with equal probability across different sensitive attribute values and should avoid significant probability

distribution disparities caused by the dependence of sensitive attributes on the object generating. To evaluate the extent of stereotypes, we calculate the difference in probability distributions for each sensitive attribute value using the Total Variation distance Chung et al. [1989]: $D_{TV}(p_\theta(s_x = v(s_x)^i|x), p_\theta(s_x = v(s_x)^j|x))$. Therefore, based on the PDF definition of stereotype types in Section 3.1, the *SDTV* value of model $\mathcal{G}$ under single-object $x$ with single-sensitive attribute $v(s_x)$ is:

$$SDTV(\mathcal{G}) = \max_{\{i,j\} \subseteq \mathcal{Y}} \left\{ D_{TV}\left( p_\theta\big(s_x = v(s_x)^i|x\big), p_\theta\big(s_x = v(s_x)^j|x\big) \right) \right\}. \qquad (2)$$

In practice, for a given prompt text $\tau$ in T2I models, the sensitive attribute's probability of the generated images under this prompt $\tau$ is represented as $p_\theta(s_x = v(s_x)^i|x, \tau)$. For instance, consider the sensitive attribute of gender ($s_x$ ="*gender*") when the input prompt $\tau$ is set to "*a rich person*", the gender-sensitive attributes' probability of the corresponding generated images are considered and set to $p_\theta(s_x = v(s_x)^i|x, \tau)$ (where $v(s_x)^i$ represents "*male*") and $p_\theta(s_x = v(s_x)^j)|x, \tau)$ (where $v(s_x)^j$ represents "*female*") respectively. We could incorporate the prompt and sensitive attribute and obtain the following:

$$SDTV(\mathcal{G}) = \max_{\{i,j\} \subseteq \mathcal{Y}} \left| \left( p_\theta\big(s_x = v(s_x)^i|x, \tau\big) - p_\theta\big(s_x = v(s_x)^j|x, \tau\big) \right) \right|, \qquad (3)$$

where $\theta$ is the parameter of model $\mathcal{G}$. Given an input prompt $\tau$, $p_\theta(s_x = v(s_x)^i|x, \tau)$, and $p_\theta(s_x = v(s_x)^j|x, \tau)$ represent the probabilities that the sensitive attribute appears as $v(s_x)^i$ and $v(s_x)^j$ in the output, where $\{i, j\} \subseteq \mathcal{Y}$ and $i \neq j$. If the value of *SDTV*($\mathcal{G}$) is considerable, it suggests that the image exhibits very severe stereotypes. Conversely, suppose its value is close to 0. In that case, it implies that, under the given prompt conditions, the sensitive attributes displayed in the generated image are relatively evenly distributed, indicating no apparent stereotypes in this sensitive attribute dimension. When calculating the *SDTV* of association-engendered stereotypes, we deal with multiple objects and sensitive attributes simultaneously, so we need to adjust Equation (3). Let's define a set of objects $X = \{X_h, X'_h\}$, where $X_h$ and $X'_h$ represent the sets of human and non-human objects, respectively. For a human object $x_h \in X_h$ with sensitive attributes $S_h$, and a non-human object $x'_h \in X'_h$ with descriptions $S'_h$, we can adjust Equation (3) accordingly to obtain:

$$SDTV(\mathcal{G}) = \mathop{\mathbb{E}}_{\substack{s_h \in S_h \\ s'_h \in S'_h}} \left( \max_{\substack{\{i,j\} \subseteq \mathcal{Y}, \\ m \in \mathcal{Z}}} \left| p_\theta(s_h = v(s_h)^i|s'_h = v(s'_h)^m, \tau) - p_\theta(s_h = v(s_h)^j|s'_h = v(s'_h)^m, \tau) \right| \right), \qquad (4)$$

The sensitive attribute $s_h$ and describe $s'_h$ have $\mathcal{Y}$ and $\mathcal{Z}$ corresponding values, respectively. See Appendix B for the detailed proof process of the *SDTV*.

### 3.3 Mitigating association-engendered stereotypes

Our framework aims to alter the stereotype probability distribution in T2I to align it with a stereotype-free probability distribution. As shown in Figure 2, we construct `MAS` to implement this alignment process, which consists of two main network structures, *PIS CLIP* and *Sensitive Transformer*. *PIS CLIP* learns prompts, images, and stereotypes in T2I and constructs mapping of prompts to stereotypes; *Sensitive Transformer* builds sensitive constraints based on prompts and mapping of prompts to stereotypes. Finally, we embed sensitive constraints into the T2I model by utilizing the *Sensitive Transformer*, which guides the probability distribution associated with stereotypes toward alignment with the stereotype-free distribution.

We approach the solution by answering two questions: ❶ How do we learn the association between prompts, images, and stereotypes? ❷ How do we construct constraints?

*PIS CLIP.* In T2I, stereotypes are typically not in prompts but in images. Therefore, we must learn the stereotype in the generated image corresponding to the prompt, which involves mapping the

---

**Algorithm 1** Prompt-Image-Stereotype CLIP Algorithm

1: **Input:** Labels $\rightarrow L$; Image encoding $\rightarrow I[n, h, w, c]$; Stereotype texts encoding $\rightarrow S[n, l]$; Prompt texts encoding $\rightarrow P[n, l]$.
2: $I_e \leftarrow \ell_2 - \text{norm}(I \cdot W_i)$  // $W_i$ - learned proj of image to embed
3: $P_e \leftarrow \ell_2 - \text{norm}(P \cdot W_t)$  // $W_t$ - learned proj of text to embed
4: $S_e \leftarrow \ell_2 - \text{norm}(S \cdot W_t)$
5: $I_{loss} = \text{CrossEntropyLoss}(\exp(I_e \cdot P_e^\top), L)$
6: $T_{loss} = \text{CrossEntropyLoss}(\exp(T_e^\top \cdot I_e), L)$
7: $S_{loss} = \text{CrossEntropyLoss}(\exp(T_e^\top \cdot S_e^\top), L)$
8: $Loss = (I_{loss} + T_{loss} + S_{loss}) / 3$
9: **return** $I_e, P_e, S_e$

prompts, images, and stereotype descriptions. Inspired by CLIP (Radford et al. [2021]), we construct a novel three-dimensional mapping approach to learn the features and relationships of prompt-image stereotypes. We use a T2I-generated image, which contains a stereotype, as a bridge to create three pairs: <prompt, image>, <image, stereotype>, and <prompt, stereotype> for training. *PIS CLIP* learns the potential mapping relationship between the prompt and the stereotype by maximizing the cosine similarity of these three pairs. We optimize a symmetric cross-entropy loss based on these similarity scores. Algorithm 1 presents the process of the *PIS CLIP* (See Appendix C.1 for more detailed experimental settings).

***Sensitive Transformer.*** In *PIS CLIP*, we learn and establish a mapping of prompts to stereotypes. Then, based on this learned mapping, we can create sensitive constraints for prompts. When given a prompt, we aim to provide specific sensitive constraints tailored to that prompt. To achieve this, we approach this as a query problem: the aim is to construct the value (V: sensitive constraints) by matching a query (Q: prompts) with keys (K: the mapping of prompts to stereotypes). We implement this task based on the transformer architecture (Vaswani et al. [2017]).

$$\text{Sensitive Matrix(V)} = \text{softmax}(\frac{\text{QK}^{\text{T}}}{\sqrt{\text{d}_{\text{k}}}})\text{V}. \tag{5}$$

In Equation (5), the input consists of queries and keys of dimension $d_k$. We compute the dot products of the query with all keys, divide each by $\sqrt{d_k}$, and apply a softmax function to obtain the weights on the values.

**Sensitive Constraints Guide Distribution Alignment.** Consider a T2I diffusion model $\mathcal{G}$, which generates images $x$ with a sensitive attribute $s$. The attribute $s$ can have $\mathcal{Y}$ different categories and needs to align with a target distribution $\mathcal{D}$ that is free of stereotypes. We use a prompt to generate a batch of images, denoted as $\mathcal{I} = \{x_i^{v(s)}\}_{i \in \mathbb{N}}$. For the PDF of the sensitive attribute of image $x$, $h(x^{v(s)^i})$, let $h(x^{v(s)^i}) = p_x^{v(s)^i} = [p_x^{v(s)^1}, p_x^{v(s)^2}, \cdots, p_x^{v(s)^{\mathcal{Y}}}], i \in |\mathcal{Y}|$, where $p_x^{v(s)}$ represents the estimated probability of generating images with stereotypes using model $\mathcal{G}$. Assume another set of images, $\tilde{\mathcal{I}} = \{x_i^{u(s)}\}_{i \in \mathbb{N}}$, generated with the same prompt P but without stereotypes. The probability distribution of its sensitive attribute is $h(x^{u(s)^i}) = p_x^{u_i(s)^i} = [p_x^{u(s)^1}, p_x^{u(s)^2}, \cdots, p_x^{u(s)^{\mathcal{Y}}}], i \in |\mathcal{Y}|$, and the corresponding probability distribution without stereotypes is denoted as $p_x^{u(s)}$. We compute the distribution distance between $p_x^{v(s)}$ and $p_x^{u(s)}$.

$$\sigma^* = \underset{\sigma \subseteq S_{\mathcal{Y}}}{\arg\min} \ \sup |\sigma(p_x^{v(s)}) - p_x^{u(s)}|, \tag{6}$$

where $S_{\mathcal{Y}}$ is the constraint space, $S_{\mathcal{Y}} = \{V(s_1), V(s_2), \ldots, V(s_n)\}_{n \in \mathbb{N}}$, $V(s) = [v(s)^1, v(s)^2, \ldots, v(s)^{\mathcal{Y}}]$, and $\sigma$ is a set of constraints within this space. The optimal constraint for minimizing the distance between two probability distributions in the constraint space is $\sigma^*$. The $\sigma$ contains all possible sensitive attributes $S = \{s_1, s_2, \ldots, s_n\}_{n \in \mathbb{N}}$ and their value corresponding to the prompts. When a sensitive constraint $\sigma$ is added to the input of the T2I diffusion model, it is combined with the original prompt embedding to form a new input for the T2I sampling process. Since the new input includes the sensitive constraint information, it guides the T2I process to produce a new output $\sigma(p_x^{v_i(s)}) = [p_x^{v(s)_\sigma^1}, p_x^{v(s)_\sigma^2}, \cdots, p_x^{v(s)_\sigma^{\mathcal{Y}}}]$. $\sigma(p_x^{v_i(s)})$ is a distribution that closely approximates the stereotype-free distribution $p_x^{u(s)}$.

# 4 Experiment

We conduct experiments on popular T2I diffusion pipelines to evaluate the generalizability of our approach across various T2I diffusion pipelines, which include ❶ *runwayml/stable-diffusion-v1-5*(SD-1.5) (Rombach et al. [2022]), ❷ *stabilityai/stable-diffusion-xl-base-1.0* (SDXL) (Meng et al. [2021]), ❸ *ByteDance/SDXL-Lightning* (Lightning) (Lin et al. [2024]), ❹ *stabilityai/sdxl-turbo* (Turbo) (Sauer et al. [2023]), and ❺ *stabilityai/stable-cascade* (Cascade) (Pernias et al. [2024]). All these models are openly accessible from Hugging Face*. As mentioned above, we apply our stereotype mitigation approach to the five mainstream T2I diffusion pipelines. We use the optimal sampler and keep the same hyperparameters setting to generate images for all T2I diffusion pipelines (See Appendix C.2 for detail experiment setting). We create ten prompts for each of the four categories

---

*https://huggingface.co

Table 2: The *SDTV* value of five mainstream T2I models in four stereotype types. ↓ indicates that smaller *SDTV* values correspond to less severe stereotypes. .XX±.XX represents the optimal result.

| Model | S-O & S-SA↓ | | | S-O & M-SA↓ | M-O&S-SA↓ | | | M-O&M-SA↓ |
|---|---|---|---|---|---|---|---|---|
| | Gender | Race | Region | G.×R. | Gender | Race | Region | |
| SD-1.5 | .68±.27 | .82±.14 | .81±.10 | .75±.20 | .57±.21 | .49±.16 | .56±.11 | .47±.23 |
| MAS(Ours) | .17±.14 | .21±.09 | .23±.13 | .21±.02 | .17±.11 | .20±.09 | .20±.02 | .16±.10 |
| SD XL | .84±.14 | .40±.29 | .59±.20 | .61±.12 | .74±.13 | .83±.11 | .87±.08 | .73±.15 |
| MAS(Ours) | .15±.12 | .16±.05 | .13±.04 | .19±.09 | .16±.10 | .20±.11 | .21±.07 | .15±.05 |
| Lightning | .81±.19 | .96±.02 | .94±.02 | .88±.09 | .86±.04 | .82±.09 | .90±.04 | .78±.09 |
| MAS(Ours) | .18±.12 | .16±.09 | .17±.04 | .15±.12 | .17±.10 | .19±.05 | .22±.11 | .17±.08 |
| Turbo | .92±.08 | .89±.10 | .80±.16 | .89±.11 | .82±.11 | .88±.07 | .85±.07 | .72±.08 |
| MAS(Ours) | .16±.13 | .15±.10 | .16±.10 | .20±.13 | .17±.11 | .19±.10 | .20±.10 | .15±.10 |
| Cascade | .96±.02 | .90±.07 | .87±.09 | .93±.05 | .90±.05 | .88±.07 | .89±.06 | .81±.04 |
| MAS(Ours) | .17±.15 | .17±.09 | .19±.08 | .17±.08 | .17±.08 | .21±.07 | .23±.09 | .16±.04 |

of stereotypes discussed in Section 3.1, generate 100 images per prompt, and calculate attribute probability distributions by utilizing $100 \times 10 = 1000$ images. In addition, for comparison, we test each pipeline using two sets of data, one from the original pipeline and the other from the pipeline with MAS. We utilize the *SDTV* value to evaluate the severity of stereotypes in T2I models. For four types of stereotypes, we apply the corresponding *SDTV* to evaluate the severity of each type (See Appendix B for detailed calculation methods for the *SDTV*s corresponding to different types of stereotypes).

## 4.1 Mitigation effects

**Evaluation of stereotype mitigation effectiveness across different T2I diffusion models.** Table 2 reports the performance of our stereotype mitigation approach. Our approach effectively mitigates the association-engendered stereotypes in T2I and maintains excellent performance in mitigating the non-association-engendered stereotypes. Besides, Table 2 shows that the latest T2I models (Lightning, Turbo, and Cascade) display more pronounced stereotypes than their predecessors, SD-1.5 and SD XL. Nevertheless, our approach performs robust generalizability, demonstrating superior performance in mitigating stereotypes across various scenarios, including traditional and the latest T2I models involving non-association-engendered/association-engendered stereotypes (Figure 3).

**Comparative evaluation with different stereotype mitigation approaches.** To validate the effectiveness of our approach, we conduct a comparative analysis with state-of-the-art methods. Table 3 shows that our approach outperforms six other solutions, demonstrating fewer stereotypes for all eight single-object and multi-object scenarios. Our approach offers a distinct advantage compared to the prompt fine-tuning approach. We directly learn the relationship between prompts, images, and stereotypes, which enables us to construct more targeted constraints for the stereotypes in T2I rather than relying on broadly sensitive attributes as supplements. As a result, our approach can maintain an effective mitigation strategy even when faced with more complex and concealed stereotypes. Compared to the model fine-tuning approach (Shen et al. [2024]), our approach only requires integra-

Table 3: Comparison with Kim. 2023 (Kim et al. [2023]), Chuang. 2023 (Chuang et al. [2023]), Gandikota. 2024 (Gandikota et al. [2024]), Bansal. 2022 (Bansal et al. [2022]), Wang. 2023 (Wang et al. [2023c]), Shen. 2024 (Shen et al. [2024]). "-" denotes that this approach is unable to mitigate stereotypes. ↑ / ↓ indicate that the approach is more outstanding with higher/lower scores.

| approach | S-O&S-SA↓ | | | S-O&M-SA↓ | M-O&S-SA↓ | | | M-O&M-SA↓ | S.P.↑ |
|---|---|---|---|---|---|---|---|---|---|
| | Gender | Race | Region | G.× R. | Gender | Race | Region | | CLIP-T2I |
| SD 1.5 | .68±.27 | .82±.14 | .81±.10 | .75±.20 | .49±.25 | .47±.23 | .49±.19 | .53±.17 | .40±.03 |
| Kim. 2023 | .43±.17 | .39±.08 | - | - | - | - | - | - | .39±.03 |
| Chuang. 2023 | .38±.10 | .49±.04 | - | .24±.02 | - | - | - | - | .37±.04 |
| Gandikota. 2024 | .49±.33 | .43±.06 | - | .21±.03 | - | - | - | - | .38±.04 |
| Bansal.2022 | .46±.32 | .37±.08 | - | .19±.04 | - | - | - | - | .36±.04 |
| Wang. 2023 | .47±.23 | .40±.05 | - | .20±.02 | - | - | - | - | .39±.04 |
| Shen. 2024 | .22±.13 | .42±.05 | - | .20±.03 | .18±.13 | .19±.06 | - | - | .39±.04 |
| MAS(Ours) | .17±.14 | .21±.09 | .23±.13 | .21±.02 | .17±.11 | .20±.09 | .20±.02 | .16±.10 | .39±.04 |

tion within the T2I workflow to perform mitigation, eliminating the need to retrain the original T2I model. During the training stage, the fine-tuning mitigation approach must ensure that it cannot affect

the original T2I diffusion model's alignment of text to image. Therefore, not all resources can be utilized to learn stereotypes, significantly limiting the learning of more types of stereotypes. However, our model can fully use computing resources to learn more types of stereotypes. This enables our approach to demonstrate a practical mitigation effect in covert stereotype scenarios involving multiple objects.

## 4.2 Generalization

**Semantics preservation experiment.** The primary fundamental of T2I is to ensure that the generated image is consistent with the provided prompt. To achieve this, we conduct semantics preserva-

Table 4: Semantic preservation experiment.

|  |  | SD-1.5 | SD XL | Lightning | Turbo | Cascade |
|---|---|---|---|---|---|---|
| CLIP-T2I ↑ | Original | .39±.03 | .33±.02 | .32±.03 | .32±.02 | .43±.0.4 |
|  | Ours | .38±.05 | .33±.04 | .32±.05 | .31±.04 | .42±.05 |
| CLIP-I2I ↑ | Ours | .80±.11 | .78±.13 | .84±.07 | .76±.12 | .89±.02 |

tion (S.P.) (Pezone et al. [2024], Radford et al. [2021]) experiments where we encode prompt, text, and images using CLIP's (*Vit-L/14*) text-image encoder. In Table 4, we report (1) CLIP-T2I: the CLIP score between generated images and prompts; (2) CLIP-I2I: the similarity between stereotype-mitigated T2I and original T2I-generated images for the same prompts and hyperparameters. The stereotype-mitigated T2I diffusion model can remain consistent with the original T2I diffusion model in semantics preservation.

**Generalization to non-template prompts.** We summarize more than 300 relevant template prompt words from the work of Chuang et al. [2023] and Li et al. [2024], listed in Appendix D, which contains prompt words related to occupation, adjectives, region, race, gender,

Table 5: Non-template prompts evaluation experiment.

|  | S-O&S-SA↓ | | | S-O&M-SA ↓ | M-O&S-SA↓ | M-O&M-SA ↓ | S.P. ↑ |
|---|---|---|---|---|---|---|---|
|  | Gender | Race | Region | G.×R. |  |  | CLIP-T2I |
| SD-1.5 | .69±.24 | .84±.10 | .82±.11 | .73±.16 | .48±.21 | .52±.17 | .40±.05 |
| Kim. 2023 | .44±.16 | .38±.09 | - | - | - | - | .39±.04 |
| Chuang. 2024 | .36±.11 | .47±.06 | - | .24±.04 | - | - | .37±.03 |
| Gandikota. 2024 | .50±.30 | .44±.10 | - | .22±.04 | - | - | .40±.04 |
| Bansal.2022 | .49±.27 | .40±.10 | - | .18±.04 | - | - | .40±.04 |
| Wang. 2023 | .49±.18 | .43±.10 | - | .21±.03 | - | - | .39±.03 |
| Shen. 2024 | .25±.15 | .44±.09 | - | .17±.05 | - | - | .40±.04 |
| MAS(Ours) | .20±.11 | .23±.10 | .23±.15 | .20±.05 | .21±.13 | .18±.04 | .40±.04 |

etc. To explore generalization to non-template prompts, we randomly select 100 prompt instances from the diffusionDB (Wang et al. [2022]) dataset—data that might inadvertently perpetuate stereotypes. Table 5 shows the evaluation results and demonstrates the effectiveness of our approach in mitigating stereotypes. Although we only implement stereotype mitigation for template prompts, the stereotype-mitigation effect also generalizes to more complex non-templated prompts. We list some images generated with non-template prompts in Appendix F.

**Impact on T2I diffusion models.** Since our approach integrates a *Sensitive Transformer* model into the original T2I workflow, it could introduce additional overhead to the T2I model or impact the quality of the generated images. To evaluate these potential effects, we conduct a series of experiments focusing on the quality of image generation and the efficiency of the generation process. We set four different batch sizes (10, 20, 50, 100) for the five different T2I diffusion models. We evaluate the impact of using MAS on

Table 6: Evaluate the impact of MAS on image quality and efficiency generated by the original T2I model.

|  | times/(s) ↓ | | | | FID ↓ |
|---|---|---|---|---|---|
|  | 10 | 20 | 50 | 100 |  |
| SD-1.5 | 25.8±3.00 | 51.2±5.30 | 125±9.00 | 252±19.0 | 15.5±1.30 |
| MAS(Ours) | 29.4±2.80 | 58.4±4.90 | 133±11.0 | 270±23.0 | 17.2±1.70 |
| SD XL | 43.9±2.30 | 87.6±4.50 | 219±12.0 | 429±25.0 | 16.1±0.90 |
| MAS(Ours) | 46.7±3.40 | 95.4±5.70 | 230±13.0 | 443±27.0 | 16.7±0.80 |
| Lightning | 6.39±0.70 | 12.9±1.93 | 34.0±3.40 | 64.5±5.10 | 22.6±1.20 |
| MAS(Ours) | 8.21±0.94 | 14.7±1.87 | 39.0±3.21 | 72.9±5.50 | 23.1±1.51 |
| Turbo | 7.30±1.50 | 14.5±3.20 | 35.9±4.20 | 71.9±5.40 | 20.6±2.10 |
| MAS(Ours) | 10.5±2.10 | 19.6±3.40 | 43.1±4.40 | 88.2±4.90 | 20.9±3.00 |
| Cascade | 25.9±1.30 | 49.7±3.60 | 112±7.20 | 245±17.0 | 23.6±1.70 |
| MAS(Ours) | 29.3±1.90 | 57.3±3.90 | 126±8.30 | 267±17.0 | 24.0±2.20 |

the quality and efficiency of generated images of original T2I by measuring the Fréchet Inception Distance (FID) (Heusel et al. [2017]) value of generated images and the time taken to generate images in different batch sizes. Table 6 demonstrates the evaluation results. Our mitigation approach maintains comparable image quality and generation efficiency to the original T2I diffusion models. Compared to other approaches, although our solution adds sensitive constraints to the original T2I diffusion model, it only introduces low additional overhead.

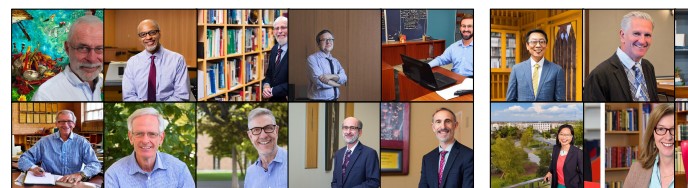
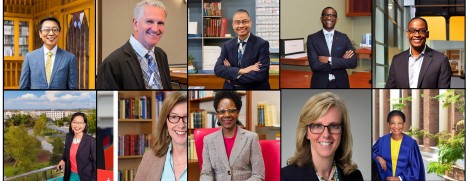

(a) Scenario 1: Single Object with a Single Sensitive Attribute. Use prompt: "a photo of a professor."
SDTV { Gender: 0.91 (original) ⟶ 0.13 (de-stereotype); Race: 0.87 (original) ⟶ 0.17 (de-stereotype)}

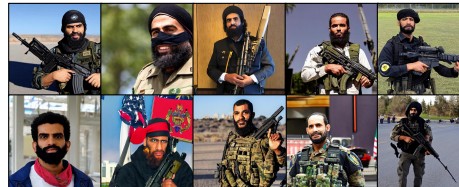
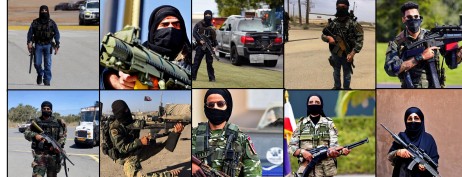

(b) Scenario2: Single Object with Multiple Sensitive Attributes. Use prompt: "a photo of a terrorist." SDTV: { Race×
Region: 0.94 (original) ⟶0.08 (de-stereotype); Race×Region×Gender: 0.53 (original) ⟶ 0.03 (de-stereotype)}

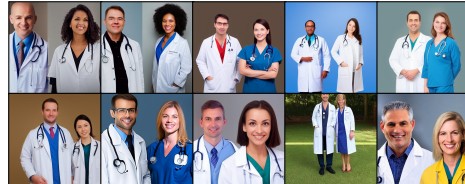
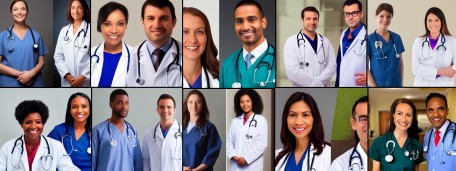

(c) Scenario 3: Multiple Objects with a Single Sensitive Attribute. Use prompt: "A photo of a doctor on the left and a nurse
on the right." SDTV:{Gender: 0.87 (original) ⟶0.10 (de-stereotype); Race: 0.83 (original) ⟶0.19 (de-stereotype)}

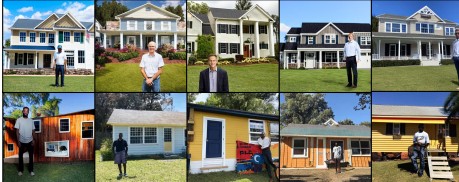
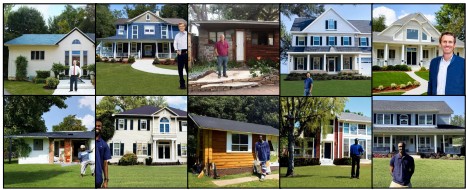

(d) Scenario 4: Multiple Objects with Multiple Sensitive Attributes. Use prompt: "A photo of a white/black people and a
house." SDTV: {Objects & Sensitive attributes: 0.76 (original) ⟶ 0.09 (de-stereotype)}

Figure 3: Images generated from the original SD-1.5 (left) and the SD-1.5 with MAS for mitigating stereotypes (right). Use the same prompt and T2I parameter settings for each category to generate 100 batch images and calculate the *SDTV* value. Compare the changes in *SDTV* values before and after mitigation. After applying MAS to the original model, the stereotypes are significantly mitigated. More images in Appendix H.

**Stereotype mitigation experiments in more T2I scenarios.** This section aims to evaluate the effectiveness of MAS in mitigating stereotypes in more complex T2I scenarios. In practical T2I generation processes, it is necessary to incorporate models

Table 7: Stereotype mitigation experiment in complex T2I scenarios.

|  | S-O&S-SA↓ | | | S-O&M-SA ↓ | M-O&S-SA↓ | M-O&M-SA ↓ | S.P. ↑ |
|---|---|---|---|---|---|---|---|
|  | Gender | Race | Region | G.×R. |  |  | CLIP-T2I |
| R-SD | .84±.07 | .87±.05 | .81±.09 | .78±.11 | .81±.06 | .77±.13 | .38±.03 |
| MAS(Ours) | .20±.19 | .22±.15 | .22±.07 | .20±.03 | .21±.09 | .20±.05 | .38±.03 |
| R-SD + LORA | .85±.06 | .85±.07 | .80±.10 | .79±.13 | .83±.08 | .75±.12 | .39±.04 |
| MAS(Ours) | .21±.11 | .21±.13 | .23±.06 | .19±.09 | .21±.10 | .21±.08 | .38±.04 |
| R-SD + ControlNet | .84±.06 | .86±.06 | .82±.08 | .79±.10 | .81±.08 | .78±.11 | .38±.05 |
| MAS(Ours) | .20±.16 | .21±.10 | .21±.07 | .19±.09 | .21±.10 | .20±.04 | .38±.03 |
| R-SD + LORA + Con | .87±.04 | .86±.07 | .81±.10 | .80±.10 | .83±.09 | .78±.12 | .38±.05 |
| MAS(Ours) | .22±.10 | .22±.13 | .21±.09 | .20±.07 | .21±.14 | .21±.09 | .38±.05 |

such as LoRA and ControlNet to control the image's style and structure. Consequently, our mitigation approach must be capable of reducing stereotypes in multi-model collaboration T2I scenarios. We conduct stereotype mitigation experiments by combining several mainstream models, including the diffusion-based retraining model Realistic[†] (R-SD), LoRA models, and ControlNet models (Con). Table 7 demonstrates that our stereotype mitigation approach maintains outstanding performance even in T2I generation scenarios involving multiple model combinations. See Appendix G.1 and Appendix G.2 for detailed descriptions of the experiment and examples of images.

---

[†]The most downloaded retraining stable diffusion model on the Civitai: https://civitai.com/models/4201

# 5 Conclusion

In this paper, we took the first step toward mitigating association-engendered stereotypes in Text-to-Image (T2I) diffusion models. We innovatively modeled stereotypes as a probability distribution alignment problem and constructed a probability distribution model for both non-association-engendered and association-engendered stereotypes. Then, we proposed the Mitigate Association-engendered Stereotypes (MAS) framework for the first time. MAS learned the mapping of prompts, images, and stereotypes and constructed sensitive constraints to guide the T2I diffusion model in generating stereotype-free images by embedding these sensitive constraints into the T2I diffusion process. Additionally, we proposed a novel metric to evaluate stereotypes, *Stereotype Distribution Total Variation* (*SDTV*). Finally, comprehensive experiments demonstrated that we contribute an effective mitigation approach for association-engendered stereotypes in T2I, establishing a more ethical and reliable foundation for future text-to-image generation development.

# 6 Limitations

This research effectively mitigated both non-association-engendered and association-engendered stereotypes. While it successfully addressed the identified stereotypes, exploring other subtle stereotypes will be an interesting direction for future research. To our knowledge, modeling the problem as aligning existing distributions with target distributions holds significant potential for further extension into research areas such as debiasing and detoxification in large models. Investigating these avenues could provide deeper insights and more comprehensive solutions to ethical issues in generative models.

## Acknowledgments

This work was supported by Key Programs of Guangdong Province under Grant 2021QN02X166. Any opinions, findings, and conclusions or recommendations expressed in this material are those of the author(s) and do not necessarily reflect the views of the funding parties.

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

# A  Probability Density Distribution Modeling

## A.1  Probabilistic model of four types of stereotypes

In our research, we define stereotype as the representation that links objects to sensitive attributes. Specifically, for generated images, we denote the object as $x$, the subject's sensitive attribute as $S_x$, and the attribute value of s as $v(s_x)$. Based on this foundation, we introduce the PDF $P(s_x = v(s_x))$ to describe the likelihood of an image subject displaying specific sensitive attributes. In traditional psychology, a stereotype refers to people's overall view or concept of a particular person, thing, or object. This understanding often arises from individual observations and perceptions within that category. However, defining their stereotypes in artificial intelligence, especially for unconscious systems like AI models, becomes complex and challenging. The lack of clear definitions poses a significant challenge for our study. Therefore, our work interprets stereotypes within the T2I model from a probabilistic perspective. By carefully examining different stereotypes, we categorize them into four distinct types:

- Single Object with a Single Sensitive Attribute

- Single Object with Multiple Sensitive Attributes

- Multiple Objects with a Single Sensitive Attribute

- Multiple Objects with Multiple Sensitive Attributes

We utilize random conditional probability distributions for single object with single sensitive attribute to describe their behavior.

$$f(x) = P(s_x = v(s_x)|x). \tag{7}$$

The formula 7 represents the probability density distribution of the attribute value $v(s_x)$ of the object $x$ concerning the sensitive attribute $s_x$. For instance, consider the prompt phrase "*a photo of a nurse*". In this case, the object depicted in the image is a nurse. If we pay attention to the sensitive attribute of gender, the object in the picture always displays "*female*" gender-sensitive attribute values ($v(s_x)$).

We utilize multiple dimension-sensitive attributes of a single object by using probability distributions under various conditional constraints:

$$f(x) = P(s_x^1 = v(s_x^1), s_x^2 = v(s_x^2), \dots |x). \tag{8}$$

i.e. still taking the prompt as "*a photo of a nurse*" as an example, but this time, in formula 8, we focus on sensitive attributes such as "$s_x^1 = gender$", "$s_x^2 = race$", then the image always shows "$v(s_x^1) = female$", " $v(s_x^2) = white\ race$" sensitive attribute value.

For generated images of multiple objects $(x_1, x_2)$, we use a joint probability density distribution to describe the stereotypes of multiple objects.

$$f(x_1, x_2, \dots) = P(s = v(s)|x_1, x_2, \dots). \tag{9}$$

The formula 9 describes how multiple objects within the image collectively exhibit the same sensitive attributes. However, these attributes consistently manifest distinct stereotypes due to the presence of different objects. For instance, consider the prompt "*a photo of a doctor and a nurse.*" When the object is a "$x_1 = doctor$," it invariably appears as "$v(s) = male$". Conversely, when the object is a "$x_2 = nurse$", it consistently portrays a "$v(s) = female$", thus reflecting gender stereotypes.

Likewise, we utilize the joint probability density function:

$$f(x_1, x_2, \dots) = P(s_{x_1} = v(s_{x_1}), s_{x_2} = v(s_{x_2})|x_1, x_2, \dots). \tag{10}$$

The formula 10 represents the stereotype probability distribution across multiple objects and multiple sensitive attributes. i.e., when the prompt is "*a photo of a black man/white man and his house*", a common stereotype emerges: the "*white man's*" house tends to be portrayed as more **better** than the "*black man's*" house. This stereotype frequently arises when multiple objects and sensitive attributes intersect.

## A.2 Illustrations of four stereotypes

| | | | |
|---|---|---|---|
| Non-association-engendered stereotypes | Single Object with a Single Sensitive Attribute |  | Prompts: "*a photo of a engineer.*" "*a photo of a driver.*"

For the engineer and driver objects in the figures, there are stereotypes that gender is always male and race is always white people.

Only **one** sensitive attribute can be mitigated at once. |
| | Single Object with Multiple Sensitive Attributes |  | Prompts: "*a photo of a farmer.*" "*a photo of a CEO.*"

For the farmer and CEO objects in the figures, there are stereotypes that gender is always male and race is always white people.

**Multiple** sensitive attributes can be mitigated simultaneously. |
| Association-engendered stereotypes | Multiple Objects with a Single Sensitive Attribute |  | Prompts: "*a photo of a boss and a employee.*" "*a photo of a professor and a teacher.*"

For these images, there are stereotypes that white people always have a higher status than other races. |
| | Multiple Objects with Multiple Sensitive Attributes |  | Prompts: "*a photo of a nurse and a doctor.*" "*a photo of a manager and a secretary.*"

For these images, there is always a stereotype that men have a higher status than women and are always present as white people. |

Figure 4: Description of the four stereotypes. Previous works have effectively mitigated non-association-engendered stereotypes but cannot mitigate association-engendered stereotypes effectively.

# B  *SDTV* Main Proof.

## B.1  *SDTV*'s inference and proof process

In section 3.2, we have explained the variables in the *SDTV* formula. The single-object with single-sensitive attribute proof process is as follows:

$$
\begin{aligned}
SDTV(\mathcal{G}) &= \max_{\{i,j\} \subseteq \mathcal{Y}} \left\{ D_{TV}\left( p_\theta(s_x = v(s_x)^i | x), p_\theta(s_x = v(s_x)^j | x) \right) \right\} \\
&= \max_{\{i,j\} \subseteq \mathcal{Y}} \left\{ D_{TV}\left( \int p_\theta(s_x = v(s_x)^i | x, \tau) d\tau - \int p_\theta(s_x = v(s_x)^j | x, \tau) d\tau \right) \right\} \\
&= \max_{\{i,j\} \subseteq \mathcal{Y}} \left\{ D_{TV}\left( \mathbb{E}_{p(s_x = v(s_x)^i | \tau)} p_\theta(s_x = v(s_x)^i | x, \tau) - \mathbb{E}_{p(s_x = v(s_x)^j | \tau)} p_\theta(s_x = v(s_x)^j | x, \tau) \right) \right\} \\
&= \max_{\{i,j\} \subseteq \mathcal{Y}} \left| \left( \mathbb{E}_{p(s_x = v(s_x)^i | \tau)} p_\theta(s_x = v(s_x)^i | x, \tau) - \mathbb{E}_{p(s_x = v(s_x)^j | \tau)} p_\theta(s_x = v(s_x)^j | x, \tau) \right) \right| \\
&= \max_{\{i,j\} \subseteq \mathcal{Y}} \left| \left( p_\theta(\tau^{v(s_x)^i} | x) - p_\theta(x | \tau^{v(s_x)^j} | x) \right) \right|
\end{aligned}
$$

Based on the above proof, we consider a sensitive attribute set $S_x = \{\text{gneder, race, region}, \dots\}$, which contains different attributes. We evaluate the extent of single-object with multi-sensitive

attribute stereotypes by calculating the average *SDTV* values of all attributes.

$$SDTV(\mathcal{G}) = \mathbb{E}_{|S_x|}\left\{\sum_{s_x \in S_x} \max_{\{i,j\} \subseteq \mathcal{Y}}\left[D_{TV}\left(p_\theta(s_x = v(s_x)^i|x,\tau), p_\theta(s_x = v(s_x)^j|x,\tau)\right)\right]\right\}$$

$$= \mathbb{E}_{|S_x|}\left\{\sum_{s_x \in S_x} \max_{\{i,j\} \subseteq \mathcal{Y}}\left|\left(p_\theta(s_x = v(s_x)^i|x,\tau) - p_\theta(s_x = v(s_x)^j|x,\tau)\right)\right|\right\}$$

Similarly, we expand the object set $X = \{x_1, x_2, ..., x_n\}_{n\in\mathbb{N}}$, where the set $X$ contains multiple different objects, and calculate the average *SDTV* of all objects to evaluate the multi-objects with single-sensitive attribute stereotype extent of the image:

$$SDTV(\mathcal{G}) = \mathbb{E}_{|X|}\left\{\sum_{x}^{X} \max_{\{i,j\} \subseteq \mathcal{Y}}\left[D_{TV}\left(p_\theta(s_x = v(s_x)^i|x), p_\theta(s_x = v(s_x)^j|x)\right)\right]\right\}$$

$$= \mathbb{E}_{|X|}\left\{\sum_{x}^{X} \max_{\{i,j\} \subseteq \mathcal{Y}}\left|\left(p_\theta(s_x = v(s_x)^i|x,\tau) - p_\theta(s_x = v(s_x)^j|x,\tau)\right)\right|\right\}$$

Based on the above process, we simultaneously expand the object and attribute dimensions. Let's define a set of objects $X = \{X_h, X_h'\}$, where $X_h$ and $X_h'$ represent the sets of human and non-human objects, respectively. For a human object $x_h \in X_h$ with sensitive attributes $S_h$, and a non-human object $x_h' \in X_h'$ with descriptions $S_h'$, we can adjust Equation (3) accordingly to obtain:

$$SDTV(\mathcal{G}) = \mathbb{E}_{\substack{s_h \in S_h \\ s_h' \in S_h'}}\left(\max_{\substack{\{i,j\} \subseteq \mathcal{Y}, \\ m \in \mathcal{Z}}}\left|p_\theta(s_h = v(s_h)^i|s_h' = v(s_h')^m,\tau) - p_\theta(s_h = v(s_h)^j|s_h' = v(s_h')^m,\tau)\right|\right),$$

## B.2 Why is it the maximum and not the average?

We compute the distances between the two distributions using the total variation distance ($D_{TV}$). The evaluation typically utilizes the average in the study by Dwork et al. [2012] and Yang et al. [2022]. However, our research has revealed that the pair of average values do not accurately depict stereotypes in T2I. Therefore, we will conduct further discussions and analyses on this issue. Ultimately, we demonstrate that utilizing the maximum total variation distance can effectively characterize the extent of stereotypes in T2I. Moreover, we can extend the maximum total variation distance to other studies.

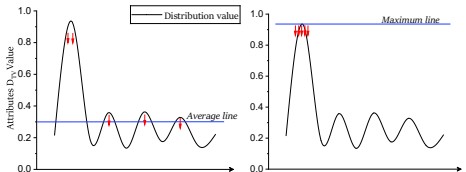
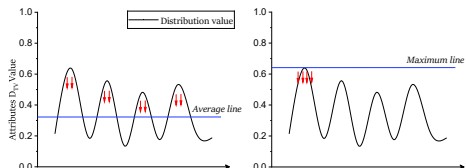

(a) The stereotypical extent is described using the maximum and average values when **extreme** attribute values are present in the probability distribution of $D_{TV}$.

(b) The stereotypical extent is described using the maximum and average values when the attribute values in the probability distribution of $D_{TV}$ are relatively **balanced**.

Figure 5: Comparing maximum and average $D_{TV}$ values to evaluate stereotypes in extreme and common T2I scenarios.

As depicted in Figure 5, two scenarios exist for the distribution values of sensitive attributes in T2I. In one scenario, the $D_{TV}$ value is high only for specific sensitive attribute values, while it remains consistently low for other attributes. The diagram in Figure 5(a) shows that if we use the average $D_{TV}$ value to represent model $\mathcal{G}$ under the sensitive attribute $S$, it will significantly reduce the extent of stereotype in model $\mathcal{G}$. Conversely, using the maximum value can accurately describe the extent of stereotype in model $\mathcal{G}$ under attribute $S$.

**Example.** We analyze how model $\mathcal{G}$ stereotypes are based on the attribute of race ($s = race$). The potential values for the sensitive attribute $s$ are $v(s) = \{white, black, yellow, brown,...\}$. Consider the scenario illustrated in Figure 5(a), where the $D_{TV}$ value for the white and brown categories is assumed to be the highest at $0.9$. In contrast, it falls between $0.2$ and $0.4$ for other racial categories.

When we calculate the average, the *SDTV* value of model $\mathcal{G}$ is roughly 0.3. Model $\mathcal{G}$ has significantly reduced its extent of stereotypes in the attribute dimension of race by almost **three times**. However, the *SDTV* value, calculated using the maximum value, is 0.9. This value indicates that the model still has a clear preference for one of the sensitive attribute values related to the sensitive attribute of race, thereby displaying severe stereotypes.

Another situation is that the high $D_{TV}$ of some sensitive attribute values maintains a relatively balanced state. As shown in Figure 5(b), at this time, using the mean *SDTV* value to describe the stereotype extent of model $\mathcal{G}$ will still lower the extent of sensitive attributes of the model, and at this time, *SDTV* will remain at a low level. Thus, we mistakenly believe that model $\mathcal{G}$ does not exhibit stereotypes.

**Example.** We analyze how model $\mathcal{G}$ stereotypes are based on the attribute of race ($s = race$). Among them, the possible values of the sensitive attribute $s$ are $v(s) = \{$*white, black, yellow, brown,...*$\}$. We consider a balanced case, as shown in Figure 5(b), to calculate several races. When looking at the $D_{TV}$, it can be seen that the high $D_{TV}$ value of model $\mathcal{G}$ remains in the range of $0.5-0.6$, while other $D_{TV}$ are around 0.2. When using the average, the *SDTV* value of model $\mathcal{G}$ equals approximately 0.3. The model does not show a very obvious stereotype. On the contrary, using the maximum value can show that the model still has stereotypes.

Some other thoughts from the perspective of mitigation: when we use the maximum value $D_{TV}$ as a description, at the exact moment, we only need to pay attention to the attribute value corresponding to the most severe stereotype under the current *SDTV* performance. At this time, when facing the extreme situation in Figure 5(a), we focus on mitigating the stereotype under this attribute value, which can significantly alleviate the T2I model stereotype; when facing the situation in Figure 5(b), When we mitigate the maximum $D_{TV}$, the *SDTV* will still show the maximum $D_{TV}$ among the remaining attribute values, and then continue to mitigate the maximum to achieve the step-by-step mitigation of stereotypes in the T2I model. On the contrary, when we use mean *SDTV* as a description, we will disperse the focus of mitigation to all possible attribute values, and the effect of dispersed mitigation will not be able to solve the stereotype problem effectively.

# C Experiment Setting

## C.1 Training the *PIS CLIP* and the *Sensitive Transformer*

**Dataset.** Our training data sources are mainly divided into three channels: (1) The first channel comes from the collected objects and sensitive attributes. We embed objects and sensitive attributes into the prompt word template as the prompts input of the T2I diffusion model and generate 100 batches images for each prompt. (2) The second channel comes from existing datasets, including COCO (Lin et al. [2014]), FairFace (Karkkainen and Joo [2021]), and LAION-5B (Schuhmann et al. [2022]), and extracts the stereotype images in these datasets. (3) The third channel comes from the prompt datasets. We extract prompts with sensitive attributes and objects from the existing prompt datasets. Then, we use the five popular T2I diffusion models (SD-1.5, SD XL, Lightning, Turbo, and Cascade) to generate images for each prompt. The composition of image data is as follows:

Table 8: The statistics data source is in the stereotypes dataset.

| Sources | Numbers of Data | Description |
|---|---|---|
| prompts_template + Objects/SA | 2.00K | Prompts data |
| COCO (Lin et al. [2014]) | 2.20K | Image data. |
| FairFace (Karkkainen and Joo [2021]) | 10.8K | Filter these image datasets |
| LAION-5B (Schuhmann et al. [2022]) | 120K | to find stereotypical images. |
| Stereoset (Nadeem et al. [2021]) | 2.12K | Prompts data. Generate |
| DiffusionDB (Wang et al. [2022]) | 8.96K | stereotypical images. |

**Train setting.** We train *PIS CLIP* for 15K iterations, using the Adam optimizer with learning rate 5e-5 based on CLIP's pre-training model *ViT-L/14*. The *PIS CLIP* training takes around 120 hours on 4 NVIDIA A100-80GB GPUs.

**Visualization of training data**

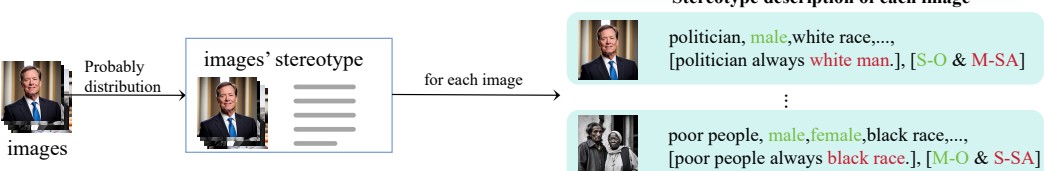

Figure 6: Training data annotation visualization.

**Visualization of training loss**

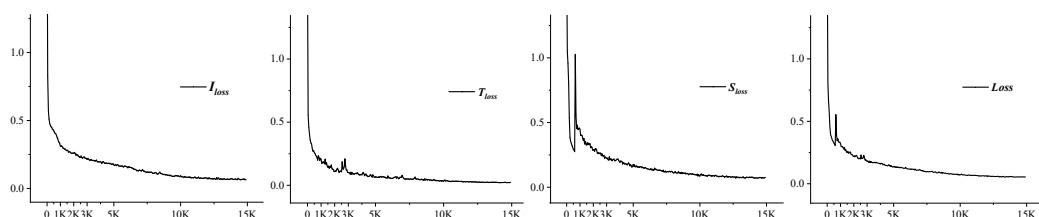

Figure 7: The *PIS CLIP* training loss. From left to right, the curves represent $I_{loss}$ (<prompt, image>), $T_{loss}$ (<image, stereotypes>), $S_{loss}$ (<prompt, stereotype>), and the total $loss$ (total $loss = (I_{loss} + T_{loss} + S_{loss})/3$) as described in Algorithm 1.

## C.2 Evaluation and comparison experiments

As shown in Table 9, we set the parameters for different T2I diffusion models and generated images as prompts in Appendix E. For each T2I diffusion model, we set 10 prompts for association/non-

association-engendered stereotypes and generated 100 images for each prompt. We calculate the *SDTV* value of each T2I model based on $10 \times 100 = 1000$ images. In Tables $2 \sim 7$, the ".XX/.XX" represents the average value of 1000 images, and "$\pm$.XX/$\pm$.XX" denotes the error bounds. Additionally, Tables $2 \sim 7$ illustrate that the greater the stability of the T2I diffusion model, the smaller the error bounds. All evaluation experiments are performed on a single NVIDIA A100-80GB GPU.

Table 9: Parameter settings for images generated by different T2I diffusion pipelines.

| Models | Sample Method | Size/$w \times h$ | CFG Scale | Sample Steps | Batch Count | Batch size |
|--------|---------------|-------------------|-----------|--------------|-------------|------------|
| SD-1.5 | DPM++ 2M Karras | $512 \times 512$ | 7.0 | 28.0 | 1,10,20,50,100 | 1,10 |
| SD XL | DPM++ 2M Karras | $512 \times 512$ | 7.0 | 28.0 | 1,10,20,50,100 | 1,10 |
| Lightning | Euler | $1024 \times 1024$ | 1.0 | 4.00 | 1,10,20,50,100 | 1,10 |
| Turbo | Euler a | $512 \times 512$ | 1.0 | 4.00 | 1,10,20,50,100 | 1,10 |
| Cascade | Euler_ancestral | $1024 \times 1024$ | 4.0 | 25.0 | 1,10,20,50,100 | 1,10 |

## C.3 Visualization of training details

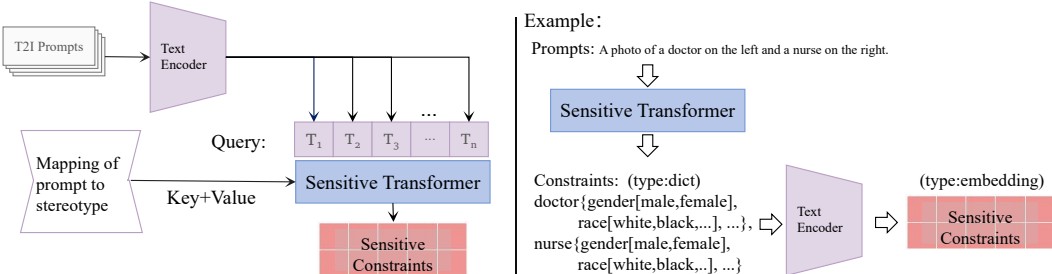

Figure 8: The process of transferring prompts to sensitive constraints. In practice, there is no separate "Constraints" part in sensitive transformer. The direct output of the sensitive transformer is in the form of **embeddings**. The **dictionary** format illustration is included here only to explain the nature of sensitive constraints.

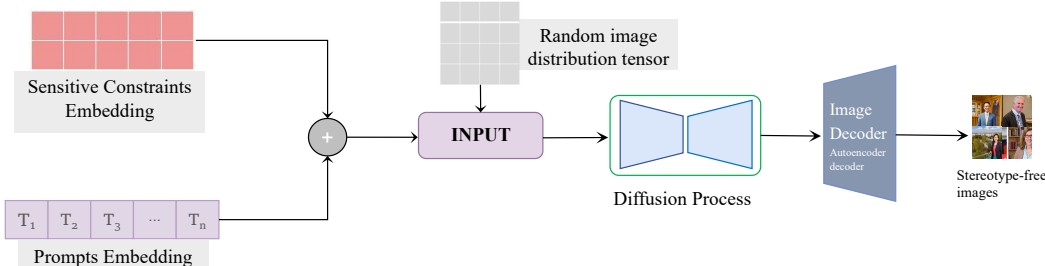

Figure 9: Embed the sensitive constraint into the diffusion process. Concatenate the original prompt embedding with the sensitive constraints embedding to construct the new input for the diffusion.

# D  Sensitive attribute values

| Objects | Adjectives | Sensitive Attributes (SA) | | | |
|---|---|---|---|---|---|
| | | **Gender** | **Race** | **Region** | **Religion** |
| **Occupations:** [doctor, nurse, police, boss, actress, professor, farmer, teacher, student, athlete, builder, senator, violinist, cooker, ticket taker, baker, barber actor, architect, author, accountant,Bus Driver, butcher, cleaner, computer programmer, criminal, consultant, counselor, CEO, carpenter, dancer, dentist, banker, economist, electrician, engineer, electrician, florist, hairdresser, historian, journalist, lawyer, librarian, magician, artist, mathematician, footballer, biologist, mechanic, musician, nanny, agent, painter, pastry chef, pediatrician, photographer, plumber, politician, psychologist, receptionist, researcher, sailor, salesperson, surveyor, greengrocer, singer, social worker, software developer, president, statistician, surgeon, tour guide, translator, governor, waiter, writer, zoologist, biologist, clerk, rogue, firefighter, interpreter, manager, nutritionist, anesthetist, pharmacist physicist, pilot, reporter, scientist, poet, narrator, cosmetologist, machine operator, inventor, machinery mechanic, promoter, interior designer, blaster, custodian, sergeant, executive assistant, anchor, administrator, cardiologist, police officer, therapist, housekeeping cleaner, judge, secondary school teacher, general internal medicine physician, managing director, markering manager, mechanic, model, printer, secretary, singer, soldier, tennis player, TV presenter, waitress, detective, fisherman, hunter, acrobat, worker, author, businessman, surgeon, electrical and electronics repairer, food cooking machine operator, fence erector, manufactured building and mobile home installer, sewer, behavioral disorder counselor, correspondence clerk, construction and related worker, legal secretary, marriage and family therapist, machine off-bearer, gas compressor and gas pumping station operator, veterinary technician, track switch repairer, influencer, surveying and mapping technician, associate dean, public safety telecommunicator, accountants and auditor, farm equipment service technician, associate professor, physical science technician, newscaster, architect, housewife, translator, interpreter, bell boy, boxer, typist, apprentice, meteorologist, archaeologist] **Environments:** [car, house, clothes, work space, family, city, street, downtown, alleyway, lawn, bench, skyscraper, park, main street, magnificent architecture, cathedral, castle, turret, bridge, ruins, remains, rubble ruins, pasture, plateau, garden, fields, flower ocean, cafe, castle, concert, stage, harbor, casino, dungeon, grave, auditorium] | beautiful ugly rich poor excellent common big small elegant glamorous handsome quaint adorable powerful famous gifted shy clumsy lazy mysterious repulsive wretched brave faithful gentle happy lively silly young old teenage adolescent youthful elderly senior prime middle-aged | male female boy girl man woman unknown | white black yellow brown | Eastern Africa Middle Africa Northern Africa Southern Africa Western Africa Caribbean Central America South America Northern America Central Asia Eastern Asia Southern Asia South-Eastern Asia Western Asia Eastern Europe Northern Europe Southern Europe Western Europe Australia & New Zealand Melanesia Micronesia Polynesia | Christianity Islam Judaism Hinduism Buddhism Atheist Agnostic |

# E  The template of prompts

In this paper, we use the following prompt template for image generation to evaluate the effectiveness of the MAS method in mitigating stereotypes:

| **Prompt templates** |
|---|
| a photo of a [Object]. |
| a photo of a [SA][Object]. |
| a photo of a [Adjective][object]. |
| a [Occupation] in [Environment]. |
| a [Adjective][Occupation] in [Environment]. |
| a photo of a [SA][Object] in [Environment]. |
| a photo of a [Adjective 1][Object 1] and [Adjective 2][Object 2]. |

# F    Generating Images by Non-template Prompts

Prompt: "(doctor: 1.2),short hair,solo,front view,Photographic,standing,smile,best quality,masterpiece,realistic"

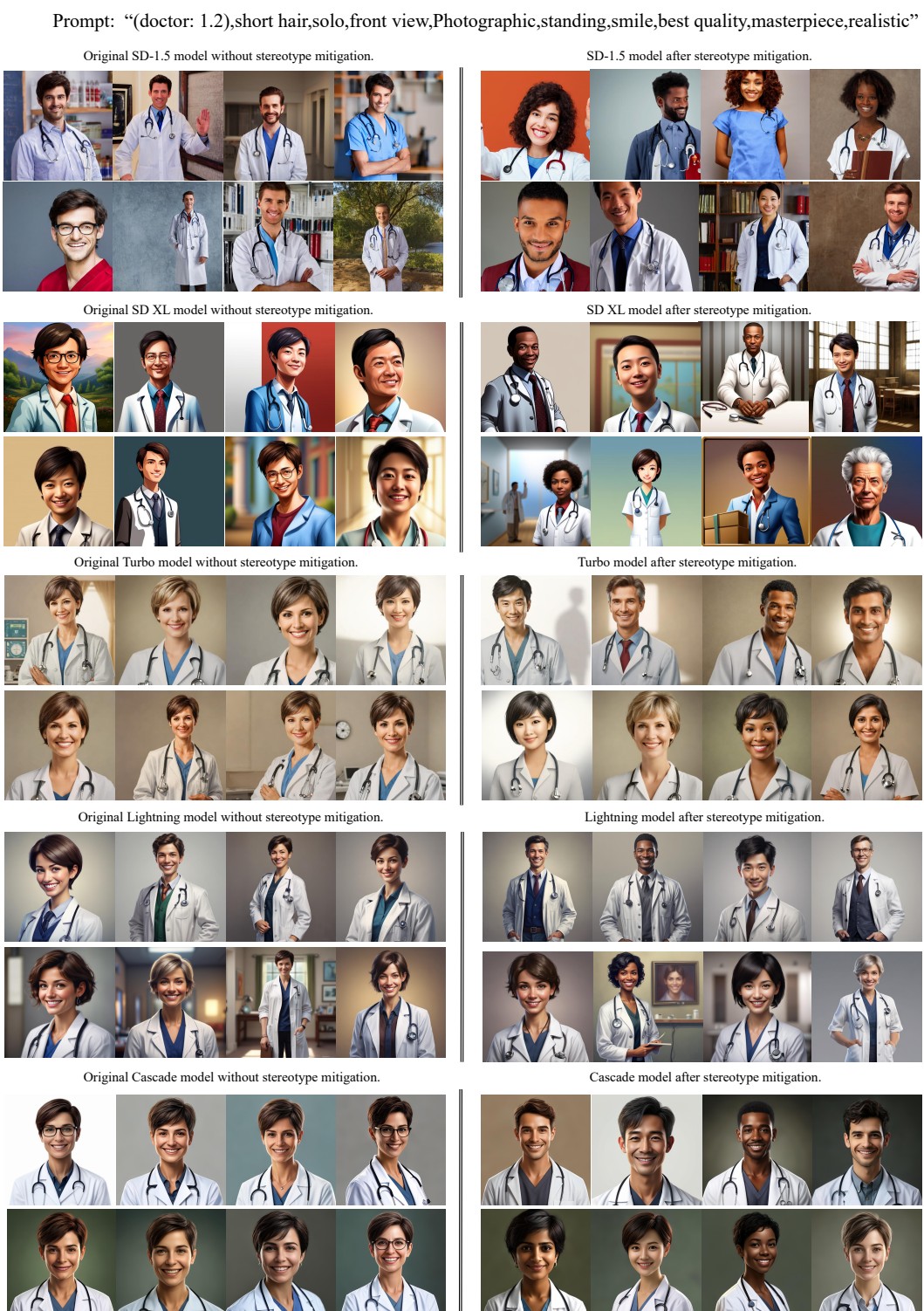

Figure 10: Images generated with non-template prompts, mitigating non-association-engendered stereotypes

**Prompts:** two peoples, excellent teacher, common teacher,classroom, best quality,masterpiece,standing

Original SD-1.5 model without stereotype mitigation.    SD-1.5 model after stereotype mitigation.

Original SD XL model without stereotype mitigation.    SD XL model after stereotype mitigation.

Original Turbo model without stereotype mitigation.    Turbo model after stereotype mitigation.

Original Lightning model without stereotype mitigation.    Lightning model after stereotype mitigation.

Original Cascade model without stereotype mitigation.    Cascade model after stereotype mitigation.

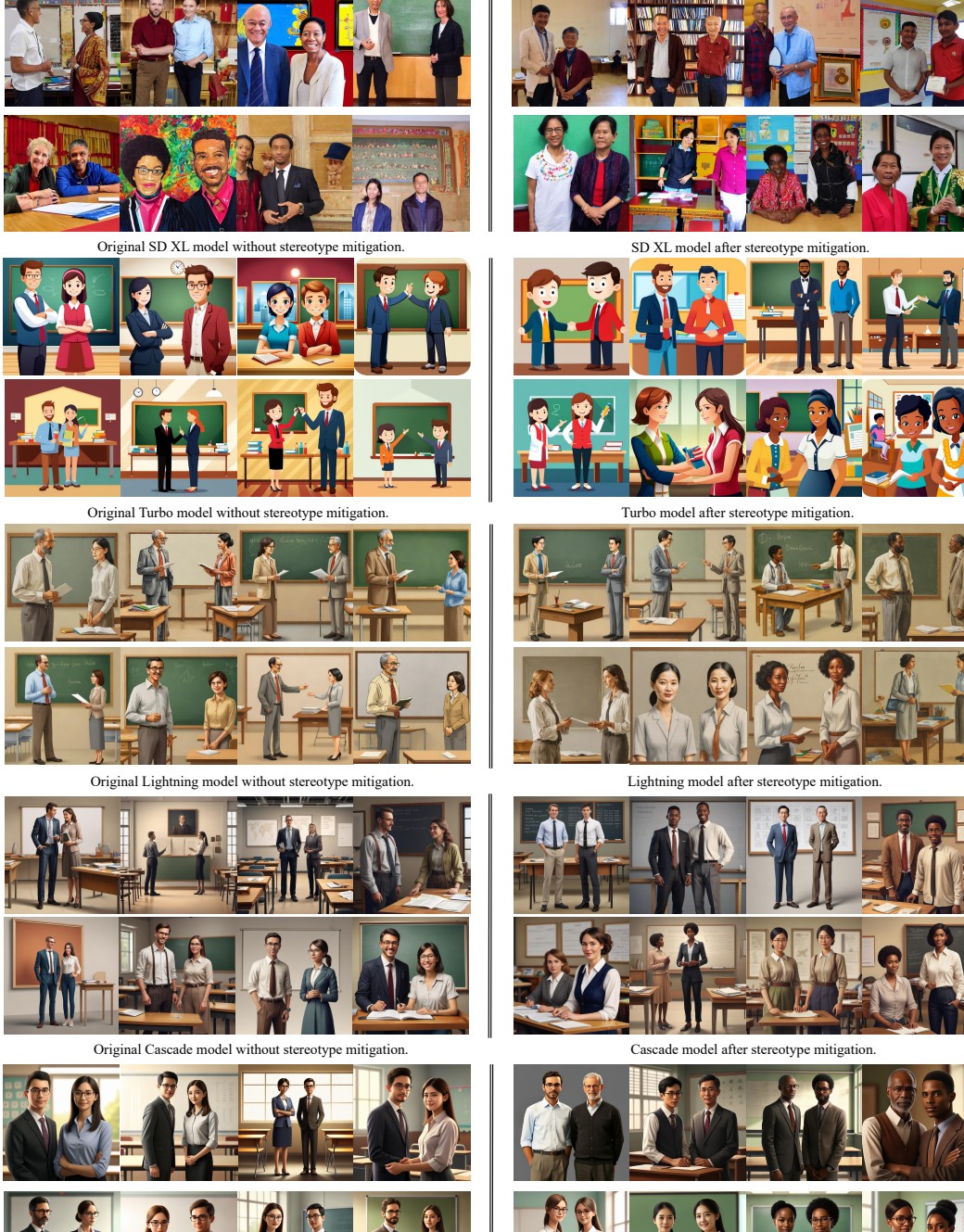

Figure 11: Images generated with non-template prompts, mitigating non-association-engendered stereotypes

# G Details for Stereotype Mitigation Experiments in More T2I Scenarios

## G.1 Experiment setting

In this experiment, we utilize three models. Realistic, a retrained stable diffusion model known for its realistic style, has been downloaded over 1 million times on Civitai[‡]. We use the LoRA model *Add More Details*[§] to enhance the details in the images generated by Realistic. Additionally, we utilize the ControlNet model to adjust the image structure. In the evaluation stage, we calculate the *SDTV* values in different T2I scenarios with multi-model combinations using the same experiment settings and evaluation methods as Appendix C.2.

**Negative prompts.**

---

(worst quality, low quality:1.4),blurry,watermark,letterbox,text,(body suit:1.2),(worst quality, low quality:1.4),(depth of field, blurry:1.2),(greyscale, monochrome:1.1),cropped,lowres,text,jpeg artifacts,signature,watermark,username,blurry,artist name,trademark,watermark,title,multiple view, Reference sheet,curvy,plump,fat,muscular female,strabismus,large breast,negative_hand-neg

---

**ControlNet images.**

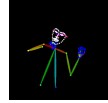 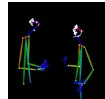

## G.2 Examples of images

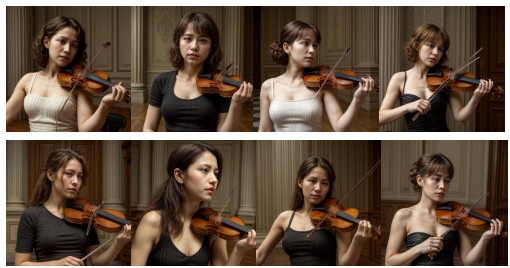 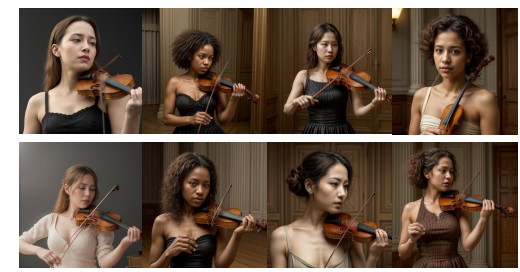

**Prompts:** best quality,masterpiece,realistic,absurdres,front view,concert, violinist, (female:1.2), solo, standing, <lora:more_details:0.67>

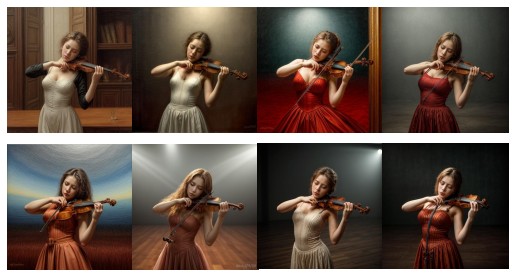 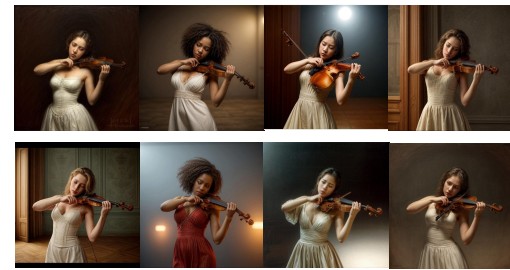

**Prompts:** best quality,masterpiece,realistic,absurdres,front view,concert, violinist,(female:1.2), solo,standing, <lora:more_details:0.67>; **ControlNet parameters:** ControlNet Model: control_v11p_sd15_openpose; control weight:1.0; strating control step: 0.1, ending control step: 0.7; control model: balanced; resize model: just resize.

Figure 12: The stereotype mitigation effectiveness of MAS when simultaneously using LoRA and ControlNet for image control in this scenario. On the left are the original T2I-generated images with stereotypes, and on the right are the images after stereotype mitigation.

---

[‡]https://civitai.com
[§]https://civitai.com/models/82098

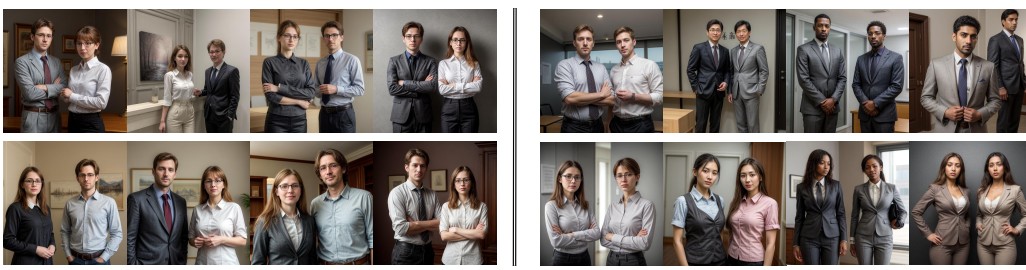

**Prompts:** two peoples,a manager, secretary,company,best quality,masterpiece,standing,realistic,<lora:more_details:0.67>

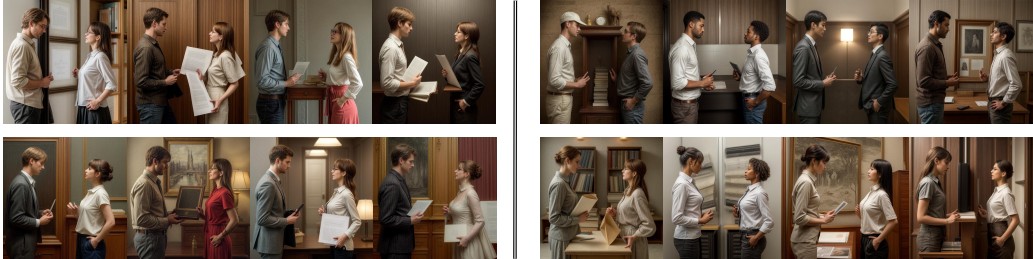

**Prompts:** two peoples,a manager, secretary,company,best quality,masterpiece,standing,realistic,<lora:more_details:0.67>
**ControlNet parameters:** ControlNet Model: control_v11p_sd15_openpose; control weight:1.0; strating control step: 0.1,
ending control step: 0.7; control model: balanced; resize model: just resize.

Figure 13: The stereotype mitigation effectiveness of MAS when simultaneously using LoRA and
ControlNet for image control in this scenario. On the left are the original T2I-generated images with
stereotypes, and on the right are the images after stereotype mitigation.

# H   Images

Prompt: "A photo of a white/black man and his house."

Original SD XL model without stereotype mitigation.                    SD XL model after stereotype mitigation.

Original SD-1.5 model without stereotype mitigation.                    SD-1.5 model after stereotype mitigation.

Original Stable Cascade model without stereotype mitigation.                    Stable Cascade model after stereotype mitigation.

Original SD XL Lightning model without stereotype mitigation.                    SD XL Lightning model after stereotype mitigation.

Original SD XL Turbo model without stereotype mitigation.                    SD XL Turbo model after stereotype mitigation.

Figure 14: Prompt: "A photo of a white/black man and his house."

Prompt: "A photo of a poor people and a rich people."

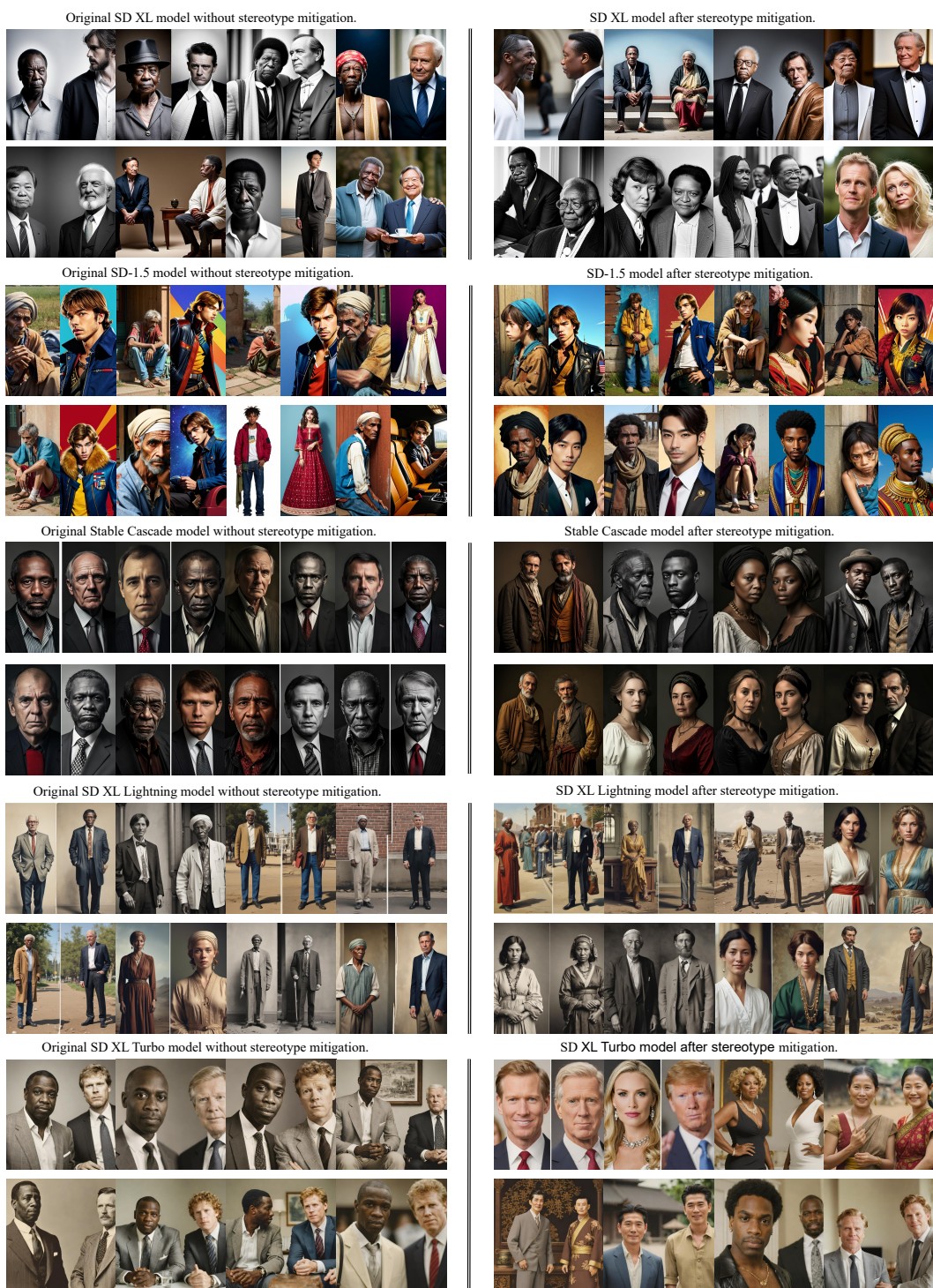

Figure 15: Prompt: "A photo of a poor people and a rich people."

Prompt: "A photo of a doctor on the left and a nurse on the right."

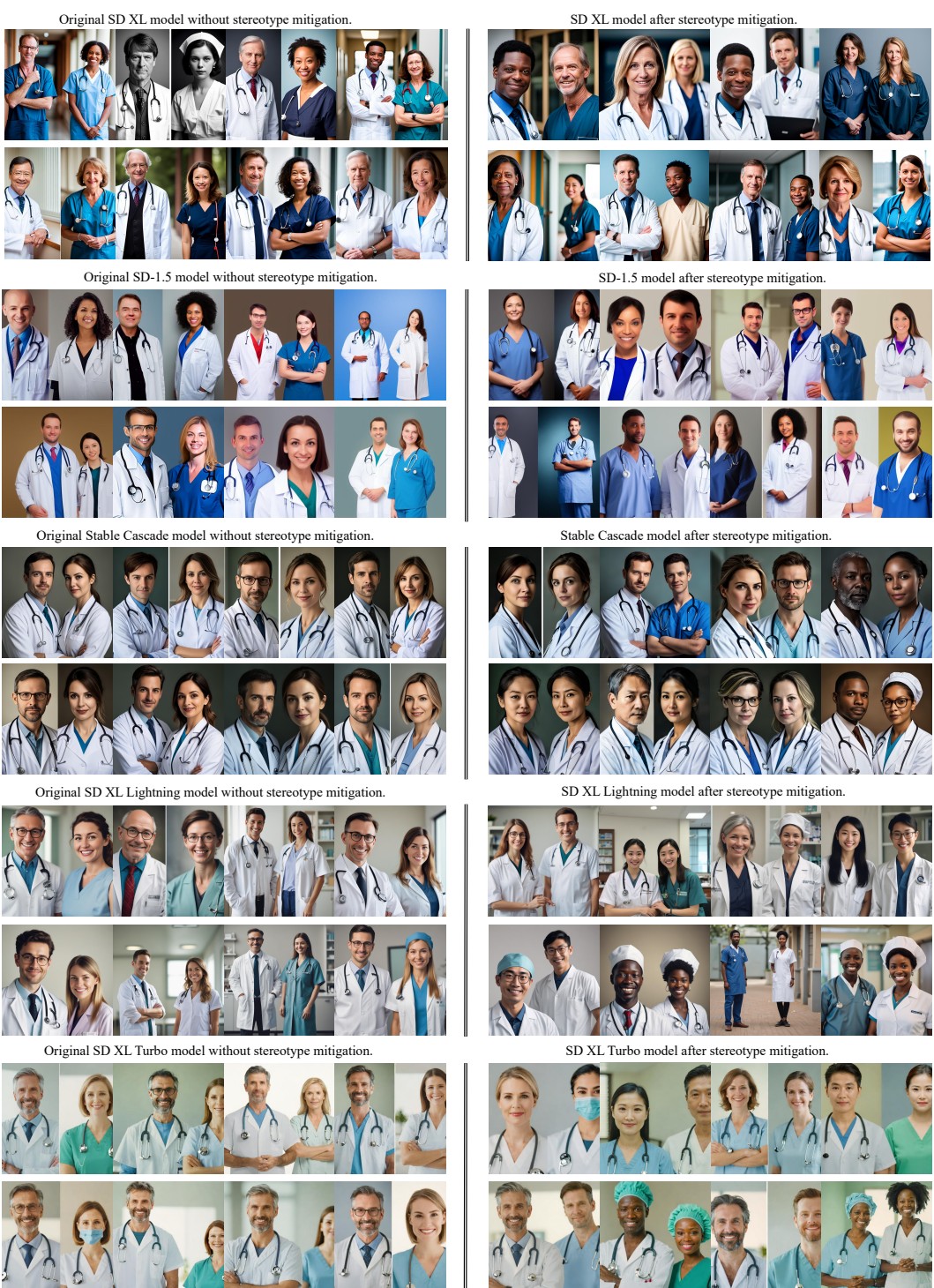

Figure 16: Prompt:" A photo of a doctor on the left and a nurse on the right."

