# OpenReview forum: "Association of Objects May Engender Stereotypes: Mitigating Association-Engendered Stereotypes in Text-to-Image Generation"
_NeurIPS.cc/2024/Conference — NeurIPS 2024 spotlight_

### Official Review · Reviewer_L6vF · 2024-07-10

**Soundness:** 3
**Presentation:** 3
**Contribution:** 3
**Rating:** 5
**Confidence:** 4

**Summary:**

The authors observe that some stereotypes only appear when an association of objects is required in T2I and propose to address association-engendered stereotypes for the first time. They use a pre-trained Text-Image-Text CLIP to map a prompt to its potential stereotypes and generate sensitive constraints using a Sensitive Transformer. The sensitive constraints are then used with the input prompt to affect T2I generation to be fairer. Besides, a new metric to evaluate association-engendered stereotypes in T2I results is proposed.

**Strengths:**

- The authors make direct observations of association-engendered stereotypes in T2I results. From these observations, the motivation to address association-engendered stereotypes is validated.
- The idea of learning mappings from the prompts to their corresponding stereotypes is interesting and the overall framework is a novel design.
- The novelly-proposed evaluation metric for association-engendered stereotypes with extensive proof and discussion is valuable.
- Extensive experiments are conducted with structural evaluations, proving the effectiveness of the method.

**Weaknesses:**

- I am wondering why the TIT CLIP can learn the mapping from prompts to stereotypes by maximizing the cosine similarity of three pairs. It is intuitively understandable that the original 2-dimensional CLIP uses contrastive learning to associate visual appearance and textual description of a single image. However, in this case, the stereotypes are something unrevealed in a single image but spotted by a statistical distribution over a set of images. The validity of pulling <prompt, stereotype> pair closer should be further explained. Besides, it would help the reader to understand the concept better if the authors could provide some examples of the stereotype descriptions used for training the TIT CLIP.
- Technical details of training the Sensitive Transformer is missing. Is it trained together with the diffusion model with the distribution alignment guidance or what is the training objective of it?
- How does the Sensitive Transformer handle engendered stereotypes across multiple attributes? Is it achieved implicitly or explicitly? Please provide further explanation.
- How to get the probability distribution for evaluation is unclear. Does it involve using pre-trained sensitive attribute classifiers like in previous works? If so, I am wondering how accurate the classifiers are when there are multiple objects in a given image.
- It would be beneficial to provide ablation studies to showcase the effectiveness of each proposed component. For example, I am curious about how a simple MLP would work in mapping the prompts to their stereotypes, compared to the TIT CLIP.
- It is visually hard to tell that the race bias is reduced in Figure 3(b), leading to concerns that the method cannot effectively mitigate biases that are strongly bonded to certain races.
- Examples after stereotype mitigation in G.2 show violation to prompt requirement "female", leading to concerns that the method might overlook the prompt input.
- One can hardly tell which person is "poor" from some examples in Figure 12. It raises concerns about how to distinguish harmful stereotypes (like relating poverty to certain races) from desirable visual concepts (like relating poverty to affordable clothes) during generations.
- The authors do not provide a discussion on the limitation of the method and its societal impact although claim to do so in the checklist.
- The stereotype mitigation target is even distribution. How can the model adapt to certain circumstances where not all attribute classes are correct? For example, the model should not generate Asian figures when asked for German soldiers in WW2.

**Questions:**

See Weaknesses for major questions.

Other minor issues include:
- L41: I think the explanation in brackets is unrelated to its previous statement. Stereotype mitigation on a single object is different from stereotype mitigation on a single sensitive attribute.
- Why there are [SA] indications in some prompt templates? If the SA is given in the prompt, the evaluation of stereotype mitigation across different SA classes is rather trivial.
- How are the sensitive constraints incorporated into the prompt input? Maybe the authors could consider redrawing Figure 2 (3) for a clearer illustration.

**Limitations:**

The authors fail to provide a discussion on limitations and potential societal impact.

---

> ### Author Rebuttal · Authors · 2024-08-05
>
> **Weaknesses (the first part)**
>
> - **W1 (TIT CLIP mapping issue):** In fact, the association-engendered stereotype in T2I can only manifest in the generated images. Therefore, we need to use images as a medium to connect the relationship between prompts and stereotypes. In our task, we consider each image's stereotype to focus on the description of the object and its sensitive attributes. Thus, each image includes two descriptions: the original 2D CLIP description of the image's meaning and the stereotype-focused description that concentrates on the object and its sensitive attributes. **As shown in Figure 1 of `supplementary_images.pdf`**, although the stereotypes in the paper are derived from probabilistic statistics, we annotate the stereotypes of the images through probabilistic descriptions rather than using the probability model for prompt-image-stereotype association mapping. Finally, in Figure 1 of **`supplementary_images.pdf`**, we provide examples of stereotype descriptions.
>
>   > Training data type:    <prompt, image, stereotype description>
>
> - **W2 (technical details):** Our Sensitive Transformer is based on Transformer[1]. As shown in Equation (5) of the paper, $\text{Sensitive Matrix}(V) = \text{softmax}\left(\frac{QK^T}{\sqrt{d_k}}\right)V$, we changed the Transformer's task from translation to generating sensitive constraints.
>
>   **The Sensitive Transformer is trained together with the diffusion model under distribution alignment guidance**.  The training objective of the Sensitive Transformer is to generate sensitive constraints that align with the given prompts. Additionally, to generate the optimal sensitive constraints for the prompt, we optimized the generated sensitive constraints. As shown in Equation (6), $\sigma^* = \mathop{\arg\min}\limits_{\sigma \subseteq S_{\mathcal{Y}}} \sup |\sigma(p_x^{v(s)}) - p_x^{u(s)}|$, the optimization objective is to minimize the total variation distance between $\sigma(p_x^{v(s)})$ and $p_x^{u(s)}$.
>
>   > [1] Ashish Vaswani, Noam Shazeer, Niki Parmar, Jakob Uszkoreit, Llion Jones, Aidan N Gomez, Łukasz Kaiser, and Illia Polosukhin. Attention is all you need. Advances in neural information processing systems, 2017, 30.
>
> - **W3 (the issue of handling multiple attributes):** **As shown in Figure 2 of `supplementary_images.pdf`**, the Sensitive Transformer generates sensitive constraints for each object in the original prompt using the prompt-image-stereotype embedding obtained from TIT CLIP. These sensitive attribute values are generated in the form of a dictionary. If the original prompt already specifies certain sensitive attributes, they will not appear in the sensitive constraints dictionary. **By simultaneously constraining multiple sensitive attributes, we suppress the association-engendered stereotypes**, achieving this in an **implicit** manner.
>
> - **W4 (the evaluation issue):** As mentioned in lines 219-221 of the paper, we provide a brief description of the evaluation of SDTV values presented in Table 2 and Table 3. The detailed calculation method for SDTV values, including single and multiple objects and sensitive attributes, is provided in Appendix B. Given the potential inaccuracy of classifiers, we evaluate the SDTV values of these images using mathematical statistical methods rather than relying on classifiers.
>
> - **W5 (the MLP suggestion):** Initially, we had the same idea as the reviewer: to use MLP to map prompts to their stereotypes. However, **as shown in Figure 1 of `supplementary_images.pdf`**, given our training data, if we skip the images and only use prompts and stereotype descriptions, we encounter the problem where the same prompt corresponds to multiple different stereotype descriptions, as illustrated in the table below.
>
>   | prompts                          | stereotype description                                       |
>   | -------------------------------- | ------------------------------------------------------------ |
>   | a photo of a doctor and a nurse. | doctor, nurse, male, female,...\[doctor always male, ...\][M-O & M-SA] |
>   | a photo of a doctor and a nurse. | doctor, nurse, female, female,...\[doctor always male, ...\][M-O & M-SA] |
>   | a photo of a doctor and a nurse. | doctor, nurse, male, male,...\[doctor always male, ...\][M-O & M-SA] |
>   | ...                              | ...                                                          |
>
>   This problem causes the MLP's loss to remain constant, preventing it from learning effective features. By encoding the image into the embedding, we can address this issue while preserving the semantic relationship between the prompt and the image, which is the main advantage of TIT CLIP over MLP.
>
> - **W6 (about Figure 3):** In Figure 3(b), the left image of the terrorist predominantly displays Middle Eastern facial features in terms of the sensitive dimension of race. After mitigation, the Middle Eastern features are covered by a mask, and the generated image then focuses on the terrorist's inherent features rather than displaying obvious racial characteristics.
>
> ---
>
> **`Note:`** Due to the word limit imposed on each rebuttal by NeurIPS, we have included our responses to **W7-W10** and the replies to the "**Questions**" in the **Author Rebuttal** cell. We **kindly request the Reviewer L6vF to switch to the Author Rebuttal cell to view the second part of our detailed responses**. We sincerely appreciate your review.

---

> > ### Comment · Reviewer_L6vF · 2024-08-12
> >
> > Thank the authors for their carefully prepared rebuttal. It addresses most of my concerns. I have raised my rating to 5.

---

### Official Review · Reviewer_E5YY · 2024-07-11

**Soundness:** 2
**Presentation:** 3
**Contribution:** 3
**Rating:** 6
**Confidence:** 3

**Summary:**

This paper presents a novel framework (MAS) to address biases in text-to-image (T2I) models. Traditional methods focus on individual object stereotypes but fail to tackle stereotypes arising from object associations. MAS models the stereotype issue as a probability distribution alignment problem, utilizing a Text-Image-Text CLIP (TIT CLIP) and a Sensitive Transformer to align generated image distributions with stereotype-free distributions. The paper also introduces the Stereotype-Distribution-Total-Variation (SDTV) metric for better stereotype evaluation. Experiments show MAS effectively mitigates both single and association-engendered stereotypes.

**Strengths:**

1. The paper addresses a relatively unexplored area in stereotype mitigation for T2I models, focusing on the association of multiple objects rather than individual objects.
2. The paper is well-structured, with a clear presentation of the problem, methodology, and results. The introduction provides a solid context for the need to address association-engendered stereotypes, and the explanation of the MAS framework is easy to follow.
3. The paper extends to improving the societal and ethical aspects of AI-generated content, making it a valuable contribution to the field.

**Weaknesses:**

1. The proposed framework adds considerable complexity to the existing T2I pipeline, which might pose challenges for practical implementation and integration into existing systems, especially for developers with limited resources or technical expertise.
2.  It is advisable to provide more details on the computational overhead introduced by the Sensitive Transformer within the MAS framework, especially in terms of real-world application performance and scalability for large-scale T2I generation tasks.
3. I wonder if the MAS framework can be adapted to other generative models beyond text-to-image, such as text-to-video or text-to-audio models.

**Questions:**

Please see the weaknesses.

**Limitations:**

Please see the weaknesses.

---

> ### Author Rebuttal · Authors · 2024-08-05
>
> **Weaknesses**
>
> - **W1 (the complexity issue):** As shown in Section 4.2 of the paper, we conducted experiments to evaluate the impact of MAS on the computational load of the T2I diffusion model. Table 6 demonstrates that MAS effectively mitigates stereotypes while maintaining image generation efficiency and quality.
>
> - **W2 (the computational overhead issue):** The time complexity of the Sensitive Transformer is consistent with that of the Transformer. For an input batch size of  $b$, sequence length $N$, and an $l$ -layer Transformer model, the computational complexity is $O(l(bNd^2 + N^2d))$, where $d$  represents the dimension of the word embeddings.
>
>   We conducted MAS tests on our local server. As reported in Table 6 of the paper, the performance is similar to the original T2I model, with a difference of about $20$ seconds over a generation task of $100$ batches with a batch size of $10$. This results in an average efficiency decrease of $0.02$ seconds per image, which is negligible for practical generation tasks. Regarding the scalability of MAS, we applied it to mainstream T2I models for stereotype mitigation. Table 2 demonstrates the effectiveness of our MAS across different stable diffusion pipelines. Additionally, our MAS implements modular integration across various pipelines, allowing stereotype mitigation by simply embedding our MAS module into the original T2I workflow.
>
> - **W3 (the adaptability issue):** Currently, our approach has only been tested for effectiveness in Text-to-Image generation. Though the concept of our approach looks feasible for Text-to-Video and Text-to-Audio generation, honestly speaking, we can not make a concrete answer about its adaptability without further deep exploration.
>
> Overall, we will carefully revise our paper based on these valuable comments.

---

> > ### Comment · Reviewer_E5YY · 2024-08-11
> > **Response to Rebuttal**
> >
> > Thank you for addressing my concerns. I decide to increase the Rating to 6.

---

### Official Review · Reviewer_ioQ9 · 2024-07-11

**Soundness:** 3
**Presentation:** 4
**Contribution:** 3
**Rating:** 8
**Confidence:** 5

**Summary:**

This paper presented the first step to mitigate association-engendered stereotypes in Text-to-Image (T2I) diffusion models. A probability distribution alignment problem was first formulated, and then a probability distribution model was constructed for non-association-engendered and association-engendered stereotypes. This paper further presented a MAS framework, which consists of the Text-Image-Text CLIP (TIT CLIP) and Sensitive Transformer. Comprehensive experiments demonstrated that the MAS framework is an effective mitigation approach for association-engendered stereotypes in T2I.

**Strengths:**

+  A novel and important research problem that tackles the stereotypes engendered by the association of multiple objects, which was ignored by previous work. This research adds critical insights into the stereotype mitigation area.

+ A neat framework MAS was proposed to mitigate such association-engendered stereotypes. MAS innovatively models the stereotype problem as a probability distribution alignment problem, which means aligning the stereotype probability distribution of the generated image with the stereotype-free distribution.

+ Two effective components, TIT CLIP and Sensitive Transformer, were proposed to enable MAS.  The framework MAS learns the mapping of prompts, images, and stereotypes via the TIT CLIP and constructs sensitive constraints via the Sensitive Transformer to guide the T2I diffusion model in generating stereotype-free images by embedding these sensitive constraints into the T2I diffusion process.

+ A novel metric, Stereotype-Distribution-Total-Variation (SDTV), was introduced to evaluate association-engendered stereotypes accurately due to the insufficiency of existing metrics.

+ Extensive experiments were conducted, supported by the 13 Appendix pages.

**Weaknesses:**

- In line 167, for the Algorithm 1, what is the output?  Please clarify.

- More output examples of the proposed framework MAS and baseline methods should be shown to illustrate the mitigation effects.

- While the paper provides an Appendix for detailed experimental settings, some implementation details may be needed in Section 4.

**Questions:**

Q1: In line 167, for the Algorithm 1, what is the output?  Please clarify.

Q2: More output examples of the proposed framework MAS and baseline methods should be shown to illustrate the mitigation effects.

Q3: In Appendix B2 Figure 5(a), what does "extreme" mean in this context?

**Limitations:**

- Please clarify the "stereotype-free distribution", which is not very clear in this paper.

- While the paper provides an Appendix for detailed experimental settings, some implementation details may be needed in Section 4.

---

> ### Author Rebuttal · Authors · 2024-08-05
>
> **Weaknesses**
>
> - **W1 (the algorithm issue):** The output of Algorithm 1 is the embedding of prompt, image, and stereotype.
> - **W2 (about the suggestions of output examples):** We sincerely appreciate your constructive suggestions. We will add examples of outputs from MAS and other baselines to the paper to demonstrate the mitigation effects alongside the results reported in Table 3.
> - **W3 (the experimental settings):** We describe the main pipeline and key evaluation methods in Section 3. Due to the paper's page limit, we have to place the detailed experimental settings in the appendix.
>
> **Questions**
>
> - **Q1:** Please see **W1**.
> - **Q2:** Please see **W2**.
>
> - **Q3 (the definition of extreme): Extreme** refers to a scenario in a sensitive attribute dimension where the occurrence probability of a specific sensitive attribute value significantly exceeds the combined occurrence probability of other attribute values. We refer to this specific sensitive attribute value as an **extreme attribute value**. For instance, when using "*an image of a beautiful woman*" as a prompt and generating some images, considering the woman's race, 90% of the images may depict the white race, while other races only account for the remaining 10%.
>
> **Clarifications**
>
> - As mentioned in lines 132-136 of the paper, stereotype-free means that a T2I model should generate images with equal probability across different sensitive attribute values, avoiding significant disparities in probability distribution caused by the dependence of sensitive attributes on the object being generated. **Stereotype-free distribution** means that the T2I model generates images with equal probability for each sensitive attribute value, resulting in a **probability distribution of sensitive attributes** close to **a uniform distribution**.
>
> - Please see **W3**.
>
> Overall, we will carefully revise our paper based on these valuable comments.

---

> > ### Comment · Reviewer_ioQ9 · 2024-08-14
> >
> > Thank you for your detailed rebuttal and the effort you have put into addressing the concerns raised. Your responses are thoughtful and show a clear understanding of the issues at hand. Based on your rebuttal, I am pleased to inform you that I am increasing my score for your submission.

---

### Official Review · Reviewer_QwVE · 2024-07-16

**Soundness:** 3
**Presentation:** 3
**Contribution:** 3
**Rating:** 7
**Confidence:** 3

**Summary:**

The paper proposes a framework to detect and mitigate stereotype association in Text-to-Image models. They conduct extensive experiments to demonstrate the usability of the framework.

**Strengths:**

+ The authors aim to assess "association-engendered" stereotypes in T2I models. They model the stereotype mitigation problem as a probability distribution alignment problem and propose MAS framework to mitigate the association-engendered stereotypes.
+ The authors conduct extensive experiments to demonstrate the effectiveness of the framework.

**Weaknesses:**

- I strongly recommend renaming the acronym for the Text-Image-Text CLIP model. The current acronym 'TIT-CLIP' is inappropriate, particularly given the project's focus on detecting stereotypes in Text-to-Image models. A more suitable name that reflects the project's serious and sensitive nature would be advisable.

- The claim that the default representation of different identity groups (e.g., 'black' and 'white' people) is not stereotypical is false (Lines 8-10, Figure 1). Research has demonstrated that the default representations of identities are indeed stereotypical, and default representations of objects can also exhibit stereotypical characteristics [1, 2, 3, 4].

[1] Easily Accessible Text-to Image Generation Amplifies Demographic Stereotypes at Large Scale. Bianchi et al. 2023 \
[2] ViSAGe: A Global-Scale Analysis of Visual Stereotypes in Text-to-Image Generation, Jha et al., 2024 \
[3] Stable bias: Evaluating Societal Representations in Diffusion Models, Luccioni et al. 2023 \
[4] AI’s Regimes of Representation: A Community-Centered Study of Text-to-Image Models in South Asia. Qadri et al., 2023 \

**Questions:**

- Since the default representation of the images can also be biased, did the authors study how much of the bias stems from the isolated representations of identity groups and objects vs the two combined?
- Can authors provide more details on the dataset used for training and evaluation and the grounding used for evaluating the presence of stereotypes before and after mitigation?

**Limitations:**

Missing limitations and discussion section.

---

> ### Author Rebuttal · Authors · 2024-08-05
>
> **Weaknesses**
>
> - **W1(the acronym issue):** We sincerely appreciate your constructive suggestions. The original acronym, Text-Image-Text CLIP, was chosen from an encoding perspective, focusing on the encoding of Text (prompt, stereotype description) and Image. We agree that this name may not be precise enough and does not fully reflect the project's theme. We will rename it to **Prompt-Image-Stereotype CLIP**. This new name better describes the objects that CLIP encodes and highlights the key aspects of the prompt and the stereotypes manifested in the generated images, making it more aligned with the project's focus than TIT-CLIP.
> - **W2 (about the default representation stereotype): We did not deny that default representations of identity are stereotypes**; therefore, we use the words "may not" in lines 8-10. Our focus in lines 8-10 and Figure 1 is on the emergence of new and subtle stereotypes when the sensitive attribute of race is associated with the object of a house, rather than on the stereotypes inherent in the representations themselves.
>
> **Questions**
>
> - **Q1 (the default stereotype issue):**  As stated in W2, the primary purpose of our paper is to demonstrate that the stereotype can be engendered by the association of two objects when these two objects individually are not stereotypical. **This association-engendered stereotype is not a simple inheritance or addition of individual default representational stereotypes but is based on the interaction between the two, engendering a new stereotype.**
> - **Q2 (the dataset issue):** We have provided a detailed description of the experimental datasets in Table 8 of Appendix C.1 of the paper. Most of the data comes from publicly available datasets, which can be accessed via the references provided in the paper.
>
> **Missing Limitations**
>
> - In lines 328-334 of the paper, we already discussed the limitations of our method and the potential social impacts.  We would extend that in the revised paper if that were not enough.
>
> Overall, we will carefully revise our paper based on these valuable comments.

---

### Author Rebuttal · Authors · 2024-08-05

**Dear Chairs and Reviewers,**

​	We kindly thank all the reviewers for their time and for providing valuable feedback on our work.

​	In response to the reviewers, we have added the **`supplementary_images.pdf`** file. This file contains annotations and explanations for the data used in our experiments and provides a more detailed description of the Sensitive Transformer.

Kind regards,

The authors

------
------

**Response to reviewer L6vF's questions (the second part):**

**Weaknesses**
- **W7 (about the Appendix G.2):** This issue arises because the "female" position in the prompts is towards the end. According to the stable diffusion encoding rules, the weight of "female" is reduced, leading to a loss of the word's features. We retested this issue, and as shown in Figure 4 of supplementary_images.pdf, adjusting the prompt weights resolves the problem. Additionally, regarding the reviewer's concerns about semantic preservation problems, we conducted semantic preservation experiments in Section 4.2 of the paper. Table 4 indicates that our MAS maintains a similar level of semantic preservation to the original T2I model.

- **W8 (about Figure 12):** In Figure 12, due to the training data of the diffusion model, the disparity between rich and poor individuals is not always very pronounced in some examples. However, an apparent phenomenon is that images of both poor and wealthy individuals predominantly depict Black and White men. This phenomenon may engender harmful stereotypes, as it suggests that being poor or wealthy is always associated with either the Black or White races. The stereotype-free T2I diffusion model should distribute the probabilities equally across all races for both poor and wealthy individuals. A wealthy person could be White, Black, Asian, etc.; similarly, a poor person should also be represented equally across all races, not just limited to White and Black men.
- **W9  (missing limitations):** In lines 328-334 of the paper, we already discussed the limitations of our method and the potential social impacts.  We would extend that in the revised paper if that were not enough.
- **W10  (the mitigation target issue):** When the prompt explicitly includes specific sensitive attributes, MAS prioritizes those specified in the prompt. As we explained in **W3**, if the original prompts already specify certain sensitive attributes, they will not appear in the sensitive constraints dictionary. For example, when generating an image of a World War II German soldier, the weight of the "German" attribute in the prompts will be significantly higher than the sensitive constraints. This ensures the model does not produce unrealistic results, such as a German soldier with Asian features.

**Questions**

- **Q1  (the Line 41 issue):** The "single object" refers to Non-Association-Engendered stereotypes. As stated in Appendix A.2, Non-Association-Engendered stereotypes include single object and single/multiple attribute cases, and the parenthetical statement aligns with Non-Association-Engendered stereotypes. We apologize for any confusion our previous wording may have caused. We will revise the sentence as follows:

  > Previous works on stereotype mitigation have been limited to a single object, referred to as *Non-Association-Engendered stereotypes* (e.g., only mitigating the stereotype problem in the occupation or gender dimension), and cannot effectively address stereotypes involving the association of multiple objects, referred to as *Association-Engendered Stereotypes*.

- **Q2 ([SA] issues in prompt word templates):** In some scenarios, the prompt already includes some sensitive attributes (SA). In such cases, we must ensure that the SA specified in the prompt remains semantically aligned while mitigating stereotypes for the unspecified SA. For example, in the prompt "a female nurse," the SA of "female" is already included. We need to maintain the "female" SA unchanged while mitigating stereotypes related to other sensitive attributes, such as the race and age of the female nurse. Therefore, when setting up the prompt template, it is crucial to consider cases where specific SA is already known.

- **Q3 (the sensitive constraints incorporated issue):** In the **`supplementary_images.pdf`**, we supplement the embedding process of the sensitive constraint in T2I, as shown in Figure 3. Using the `np.concatenate` function, we concatenate the original prompt embedding with the generated sensitive constraints embedding to form the input for the diffusion process.

**Missing Limitations**

- In lines 328-334 of the paper, we already discussed the limitations of our method and the potential social impacts. We would extend that in the revised paper if that were not enough.

Overall, we will carefully revise our paper based on these valuable comments.

---

### Decision · Program_Chairs · 2024-09-25

**Decision:**

Accept (spotlight)

**Comment:**

The paper studies the problem of association biases in text to image generation that occurs when the text prompt includes both people and objects. All of the reviewers found the paper proposed a novel method for an important problem and the evaluation was solid.